# Sex differences in BNST signaling and BNST CRF in fear processing

Olivia J Hon[1], Sofia Neira[1], Meghan E Flanigan[1], Alison V Roland[1], Christina M Caira[1], Tori Sides[1], Shannon L D'Ambrosio[1], Sophia I Lee[1], Yolanda Simpson[1], Michelle C Buccini[1], Samantha Machinski[1], Waylin Yu[2], Kristen M Boyt[1], Thomas L Kash[1,3]*

[1]Bowles Center for Alcohol Studies, University of North Carolina School of Medicine, Chapel Hill, United States; [2]Inscopix, Mountain View, United States; [3]Department of Pharmacology, University of North Carolina School of Medicine, Chapel Hill, United States

## eLife Assessment

This **valuable** study advances understanding of how corticotrophin releasing factor in the bed nucleus of the stria terminalis regulates sustained and phasic fear and how this differs between sexes. The evidence is **convincing** and based on state-of-the-art techniques. The work will be of interest to neuroscientists studying the biological basis of fear processing.

**\*For correspondence:**
tkash@email.unc.edu

**Abstract** Fear responses to perceived danger are critical for survival, as they prompt the individual to respond to threats and avoid harm. However, excessive fear can impede normal biological processes and become harmful. This study investigates the neural mechanisms underlying two distinct forms of fear—phasic and sustained—in male and female mice, with a focus on the bed nucleus of the stria terminalis (BNST) and corticotropin-releasing factor (CRF) signaling. Phasic fear is characterized by immediate responses to clear threats, while sustained fear is driven by ambiguous or uncertain cues and persists longer. Using rodent models, we found that sustained fear, modeled by partial fear conditioning, induced greater arousal and BNST activity in males, especially during ambiguous threat cues. In contrast, females exhibited reduced BNST and BNST[CRF] activity, highlighting significant sex differences in fear learning and expression. Additionally, CRF is crucial for appropriate fear response in females, as CRF knockdown led to increased fear responses, but had no effect in males. These sex-specific differences could help inform the development of targeted treatments for anxiety and trauma-related disorders, which disproportionately affect women.

## Introduction

Fear is a protective response to perceived danger that allows an organism to identify and respond to threats to avoid harm. While fear is essential for survival, excessive fear can interfere with crucial biological processes (*Barrett, 2015*; *Blanchard et al., 2011*; *Fanselow and Lester, 1988*; *Kavaliers and Choleris, 2001*; *Mobbs et al., 2015*), emphasizing the need for accurate risk assessment to maintain overall well-being. In preclinical research, risk assessment is often measured through cue-danger associations in fear conditioning paradigms, where a tone cue predicts an aversive event. The resulting fear response to the tone is used to assess fear learning. Fear conditioning paradigms can vary in how cue-danger associations are paired, leading to different behavioral states underpinned by distinct neural mechanisms. For instance, phasic fear is considered an adaptive response and is characterized by a reaction to a clear and discrete cue that dissipates rapidly once the threat is no

longer present. In rodents, this is often modeled using a fully reinforced cued-fear paradigm, where a tone exposure always co-terminates with a foot shock, thus becoming a discrete cue for this threat. Conversely, sustained fear is a heightened state of arousal and anxiety that is not clearly associated with specific cues and lasts for longer periods of time (*Davis et al., 1997*; *Davis et al., 2010*). Rodent models for this include a partially reinforced cued-fear paradigm, where tones do not always co-terminate with shock and, therefore, become uncertain cues for danger (*Glover et al., 2020*).

Most fear learning studies have been conducted in male rodents and reveal that the basolateral and central amygdala play key roles in mediating fear responses. These brain regions are activated during exposure to the tone and the subsequent fear reaction, and their inhibition disrupts normal fear processing. While the BNST, part of the extended amygdala, is not always included in canonical fear circuitry, its recruitment is particularly crucial when there is uncertainty regarding when or if harm will occur and is thought to drive sustained fear (*Davis et al., 2010*; *Glover et al., 2020*; *Goode and Maren, 2017*; *Goode et al., 2019*; *Urien and Bauer, 2022*; *Naaz et al., 2019*). However, the precise role that the BNST plays in uncertain fear processing remains unclear, particularly in females. The CRF system in the BNST is also implicated in sustained fear. CRF infusions to the BNST enhance sustained fear and emotional memory recall (*Liang et al., 2001*), while CRF receptor antagonism in the BNST blocks sustained fear (*Davis et al., 2010*; *Walker et al., 2009*). Interestingly, overexpression of CRF in BNST-CRF-expressing (BNST$^{CRF}$) neurons prior to conditioning weakens sustained fear (*Sink et al., 2013*). These findings suggest that BNST$^{CRF}$ neurons and CRF inputs to the BNST are both important for fear expression, though the precise mechanisms by which they influence fear remain unclear.

Critically, though mood disorders are more prevalent in women and non-binary individuals, preclinical studies have classically only included males (*McLean et al., 2011*; *Thorne et al., 2019*). Sex differences in defensive behavioral strategies, BNST structure and function, and CRF signaling have been reported, and thus, investigation into how these features shape negative emotional states is prudent (*del Abril et al., 1987*; *Levine et al., 2021*; *Shansky, 2015*; *Lebron-Milad and Milad, 2012*; *Bangasser and Wiersielis, 2018*; *Babb et al., 2013*; *Janitzky et al., 2014*; *Salvatore et al., 2018*). Studies suggest that females have more BNST$^{CRF}$ neurons (*Chudoba and Dabrowska, 2023*; *Uchida et al., 2019*) in this sexually dimorphic region, which may drive innate sex differences in fear learning and expression. In this study, we directly examine the contribution of BNST$^{CRF}$ signaling to phasic and sustained fear in male and female mice to test the overarching hypothesis that plasticity in BNST$^{CRF}$ neurons drive distinct behavioral responses to unpredictable threat in males and females. Since sustained fear paradigms better capture the symptoms of human anxiety, understanding the neural correlates that distinguish phasic and sustained fear is critical for improving treatments for anxiety disorders (*Davis et al., 2010*).

## Results
### Partially reinforced fear conditioning drives hyperarousal in males

To characterize the behavioral profiles and underlying neural processing that distinguish phasic and sustained fear, we subjected male and female C57BL/6J mice to either partially reinforced fear conditioning (Part), fully reinforced fear conditioning (Full), or tone-only exposure (Ctrl). To model phasic fear, we used a fully reinforced fear paradigm, where mice are exposed to three tone-shock pairings such that 100% of tones are paired with a shock. To model sustained fear, we used a partially reinforced fear paradigm where mice are presented with six tones, where only the first, second, and fourth tones are paired with a shock (*Glover et al., 2020*). Twenty-four hours after fear conditioning, all mice were subjected to a fear recall protocol consisting of six tone presentations (*Figure 1A*). Freezing was scored as a measure of defensive behavior.

As expected, the tone-only control mice displayed minimal freezing behavior during conditioning, and, while freezing increased over time, there was no significant differences between male and female mice during either Full or Part fear acquisition (*Figure 1B*). When comparing freezing to all tones, we find that freezing is significantly different between Ctrl, Full, and Part fear mice, but not between males and females (*Figure 1C*). Because freezing increases over time, it is possible that these differences are driven by differences in paradigm length. Therefore, we compared freezing to the last tone shock pairing (tone 3 in Full fear and tone 4 in Part fear) and found no differences between Full and

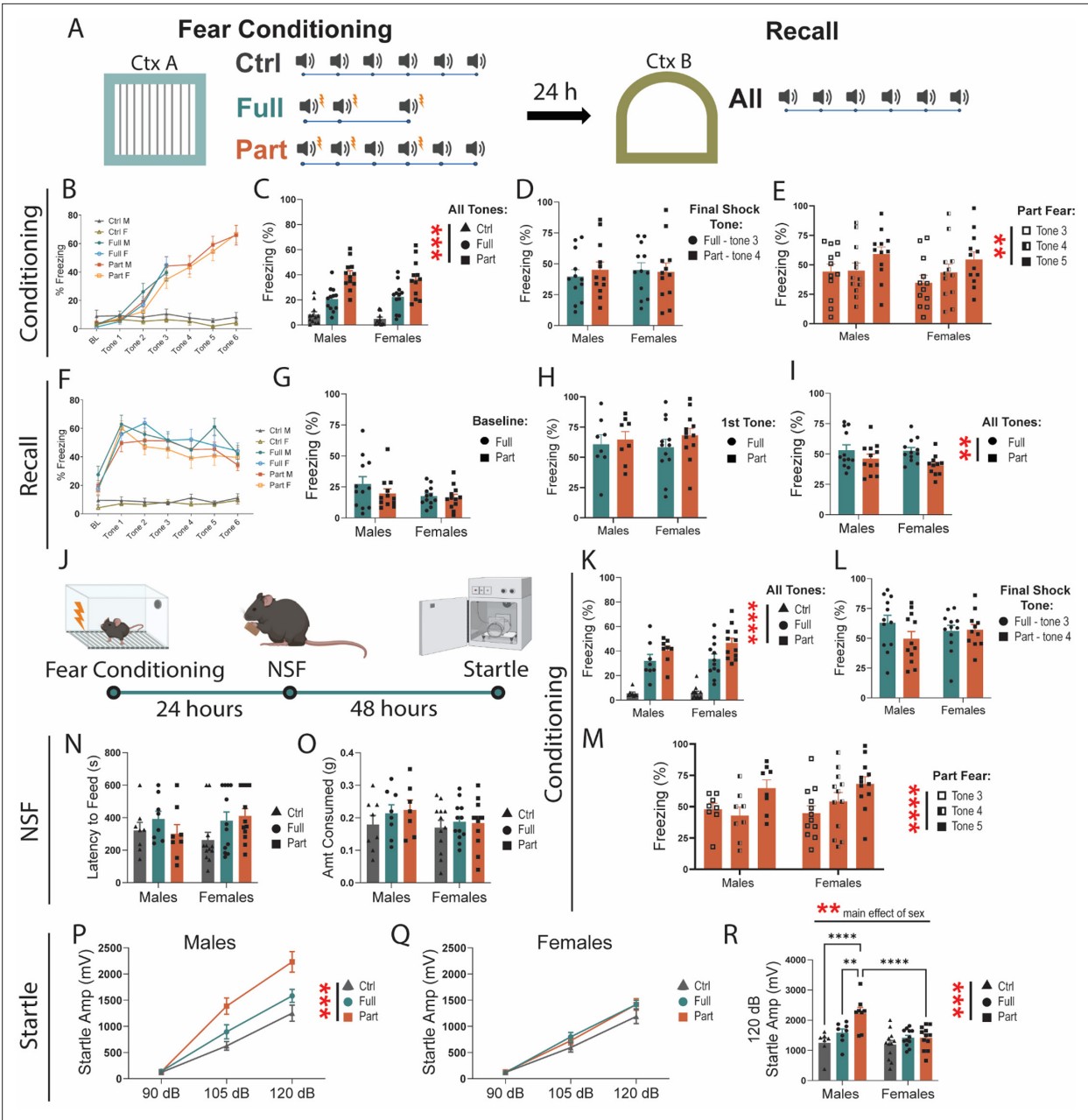

**Figure 1.** Partially reinforced fear drives a hyperarousal phenotype in males. (**A**) Schematic of behavior chambers and tone/shock presentations for fully- and partially reinforced fear conditioning and recall. (**B**) Freezing during baseline and 0–28 s of tone presentations in Ctrl (two-way RM ANOVA, main effect of tone: $F_{(3.137, 69.01)}=1.449$ ns, sex: $F_{(1, 22)}=1.717$ ns, tone × sex: $F_{(6, 132)}=0.2104$ ns), Full (two-way RM ANOVA, main effect of tone: $F_{(2.046, 45.01)}=45.19$, $p<.0001$, sex: $F_{(1, 22)}=0.4527$ ns, tone × sex: $F_{(3, 66)}=1.282$ ns), and Part (two-way RM ANOVA, main effect of tone: $F_{(3.218, 70.79)}=70.73$, $p<0.0001$, sex: $F_{(1, 22)}=0.3841$ ns, tone × sex: $F_{(6, 132)}=0.4960$ ns) fear. (**C**) Freezing for all tones from 0 to 28 s during fear conditioning (two-way ANOVA, main effect of fear: $F_{(2, 66)}=53.06$, $p<0.0001$, sex: $F_{(1, 66)}=0.4488$ ns, fear × sex: $F_{(2, 66)}=0.4588$ ns). (**D**) Full tone 3 and Part tone 4 freezing during fear conditioning (two-way ANOVA, main effect of fear: $F_{(1, 44)}=0.1043$ ns, sex: $F_{(1, 44)}=0.08787$ ns, fear × sex: $F_{(1, 44)}=0.3009$ ns). (**E**) Part fear tone 3, 4, 5 freezing during conditioning (two-way RM ANOVA, main effect of tone: $F_{(2, 44)}=6.578$, $p<0.01$, sex: $F_{(1, 22)}=0.5318$ ns, tone × sex: $F_{(2, 44)}=0.3213$ ns), (**F**) Freezing during recall baseline and 0–28 s of tone presentations in Ctrl (two-way RM ANOVA, main effect of tone: $F_{(3.394, 74.66)}=1.449$ ns, sex: $F_{(1, 22)}=1.261$ ns, tone × sex: $F_{(6, 132)}=0.6956$ ns), Full (two-way RM ANOVA, main effect of tone: $F_{(4.980, 109.6)}=12.19$, $p<0.0001$, sex: $F_{(1,22)}=0.1797$ ns, tone × sex: $F_{(6, 132)}=1.274$ ns), and Part (two-way RM ANOVA, main effect of tone: $F_{(4.754, 104.6)}=14.43$, $p<0.0001$, sex: $F_{(1, 22)}=0.0483$ ns, tone × sex: $F_{(6, 132)}=1.065$ ns) fear. (**G**) Freezing during baseline period in recall Full and Part fear (two-way ANOVA, main effect of fear: $F_{(1, 43)}=2.732$ ns, sex: $F_{(1, 43)}=1.876$ ns, fear × sex: $F_{(1, 43)}=1.776$ ns). One male Part fear mice excluded as outlier. (**H**) Freezing during recall first tone exposure (two-way ANOVA, main effect of fear: $F_{(1, 43)}=1.228$ ns, sex: $F_{(1, 43)}=0.0021$ ns, fear × sex: $F_{(1, 43)}=1.687$ ns) 1 female Part fear mouse excluded as outlier. (**I**) Average freezing for all tone presentations in Recall Full and Part Fear (two-way ANOVA, main effect of fear: $F_{(1, 43)}=7.401$ $p<.01$,

*Figure 1 continued on next page*

*Figure 1 continued*

sex: F(1,43)=0.7413 ns, fear × sex: F(1, 43)=0.5288 ns) one female Part fear mouse excluded as outlier.(J) Behavioral timeline for Novelty-Induced-Suppression-of-Feeding (NSF) anxiety-like and Acoustic Startle arousal behavioral tests after fear conditioning. (K) Freezing for all tones from 0 to 28 s during fear conditioning (two-way ANOVA, main effect of fear: F(2, 54)=57.10, p<0.0001, sex: F(1, 54)=0.7934 ns, fear × sex: F(2, 54)=0.3522 ns). (L) Full tone 3 and Part tone 4 freezing during fear conditioning (two-way ANOVA, main effect of fear: F(1, 36)=2.236 ns, sex: F(1, 36)=0.5574 ns, fear × sex: F(1, 36)=0.3593 ns). (M) Part fear tone 3, 4, 5 freezing during conditioning (two-way RM ANOVA, main effect of tone: F(2, 36)=14.53, p<0.0001, sex: F(1, 18)=0.2412 ns, tone × sex: F(2, 36)=1.541 ns), (N) Latency to feed in the novelty-induced suppression of feeding (NSF). (O) Food consumed during NSF refeed test. (P) Acoustic startle response in males (two-way RM ANOVA, main effect of fear F(2, 20)=11.61, *p*=0.0005, main effect of stim intensity: F(1.808, 36.15)=166.9 p<0.0001, interaction: F(4,40)=6.096, *p*=0.0006, Tukey post-hoc: 120 dB Full v. Part, *p*=0.0395). One Ctrl 90 dB male excluded as outlier. (Q) Acoustic startle response in females (two-way RM ANOVA, main effect of fear F(2, 33)=1.869 ns, main effect of stim intensity F(1.7452, 57.57)=223.1, p<0.0001). n=8–12/group. (R) 120 dB acoustic startle response in males and females (two-way RM ANOVA, main effect of fear F(2, 53)=10.01, p<0.001, main effect of sex F(1, 53)=9.951, p<0.01, fear × sex: F(2, 53)=4.623, p<0.05: Tukey post-hoc Males Ctrl vs. Part *p*<0.0001, Males Full vs. Part *p*<0.01, Part Males vs. Females *p*<0.0001) 1 Male 90 dB Ctrl mice excluded as outlier. n=8–12/group. All data are shown as mean + SEM. *=p<0.05, **=p<0.01, ***=p<0.001, ****=p<0.0001.

The online version of this article includes the following source data for figure 1:

**Source data 1.** This source data contains all of the data from the graphs in this figure.

Part fear mice (*Figure 1D*). This suggests that freezing differences between Full and Part fear conditioning are a product of Part fear consisting of more tones.

The Part Fear paradigm allows for careful dissection of uncertain fear learning by allowing for within-mouse comparisons before and after uncertainty are introduced. As mice are shocked after tones 1 and 2, the third tone serves as a danger signal. However, as tone 3 does not end in shock, the fourth tone becomes an uncertain predictor of danger. Tone 4 again co-terminates with a shock, introducing further uncertainty. To understand how an ambiguous threat impacts behavior, we compared freezing during tones 3, 4, and 5 in the Part fear group. Interestingly, regardless of sex, mice froze more during the 5th tone (*Figure 1E*). This is unlikely due to a gradual increase in freezing over time, as tone 3 and 4 responses are indistinguishable.

During recall, Ctrl mice froze very little, and there were no differences in freezing between sexes for Ctrl, Full, or Part Fear (*Figure 1F*). There were also no differences in freezing between Full and Part mice during the baseline period (*Figure 1G*), however, similarly to what *Glover et al., 2020* found, Part fear froze slightly less than Full fear mice during tone presentations, regardless of sex (*Figure 1I*).

Studies in rodents and humans have shown that paradigms in which aversive stimuli are uncertain or temporally unpredictable drive increased arousal and avoidance behaviors (*Glover et al., 2020*; *Goode et al., 2019*; *Urien and Bauer, 2022*; *Grillon et al., 2004*; *Davis and Walker, 2014*). We subjected Ctrl, Full, and Part fear mice to conditioning as described above, followed by a novelty-induced suppression of feeding (NSF) test 24 hr after and an acoustic startle reflex test 48 hr after, to assess how fear drives elements of avoidance behaviors and arousal, respectively. Again, we found that during conditioning, Part fear mice froze more in the final tone than preceding tones (*Figure 1K*), but differences in freezing between Part and Full fear were not observed (*Figure 1L*). There were no differences between fear conditions or sexes in latency to feed (*Figure 1N*) or amount consumed during the refeed portion (*Figure 1O*) of the NSF test. In the acoustic startle test, males exhibited potentiated startle response after both Full and Part fear conditioning (*Figure 1P*) while females did not (*Figure 1Q*). Notably, at the highest sound intensity, Part fear males showed a larger startle response than Full fear males (*Figure 1R*), suggesting that Part fear conditioning drives hyperarousal in males.

## BNST dynamics differ during part and full fear

To characterize BNST dynamics, we injected a viral construct containing the genetically encoded calcium sensor GCaMP6s into the BNST of C57BL/6J mice, implanted fiber optic cannulae for fiber photometry, and recorded calcium activity during Full and Part Fear conditioning and recall (*Figure 2A*). Given that the BNST is recruited in unpredictable fear paradigms (*Davis et al., 2010*; *Glover et al., 2020*; *Goode et al., 2019*; *Urien and Bauer, 2022*), we hypothesized that the BNST would be recruited during both Full and Part fear conditioning, but show more activity in Part fear.

First, we compared BNST responses to tone onset (first 2 s). Considering all tones, Part fear conditioning led to a greater tone response than Full fear, which post hoc testing suggests is driven by the

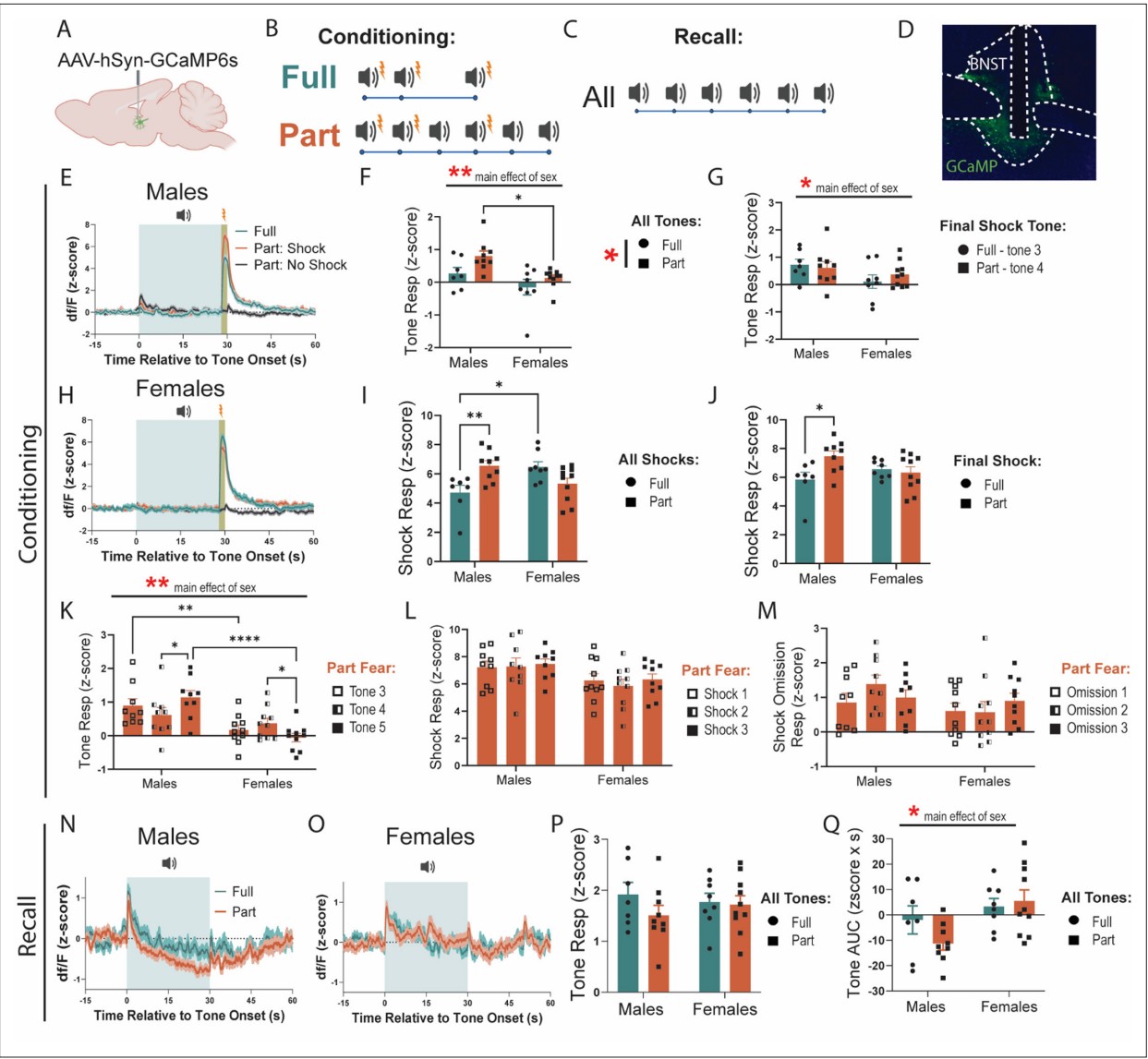

**Figure 2.** Bed nucleus of the stria terminalis (BNST) is dynamically engaged during fear conditioning and recall. (**A**) Surgical schematic for fiber photometry experiments. (**B, C**) Schematic of tone and shock presentations during fear conditioning and recall. (**D**) BNST representative images showing fiber placement. (**E**) Fear Conditioning traces in males. (**F**) BNST response to tone onset (0–2 s) for all tones (two-way ANOVA, main effect of fear: F(1, 30)=5.834, p<0.05, sex: F(1, 30)=10.56, p<0.01, fear × sex: F(1, 30)=0.5614 ns; Bonferroni's post-hoc test: Part Males vs Females p=0.0105). (**G**) Comparison of Full tone 3 and Part tone 4 BNST response (two-way ANOVA, main effect of fear: F(1, 30)=0.1296 ns, sex: F(1, 30)=4.216, p<0.05, fear × sex: F(1, 30)=0.8026 ns). (**H**) Fear Conditioning traces in females. (**I**) Shock response in Full and Part fear conditioning averaged from t=28–30 across all shock trials (two-way ANOVA, main effect of fear: F(1, 30)=0.7701 ns, sex: F(1, 30)=0.4407 ns, fear × sex: F(1, 30)=14.11, p<0.001, Bonferroni's post-hoc: Males Full vs Part p=0.0068, Full Males vs Females p=0.0121). (**J**) Final shock response in Full and Part fear conditioning averaged from t=28–30 (two-way ANOVA, main effect of fear: F(1, 30)=3.142 ns, p=0.09, sex: F(1, 30)=0.2762 ns, fear × sex: F(1, 30)=5.682, p<0.05, Bonferroni's post-hoc: Males Full vs Part p=0.0155). (**K**) BNST Tone response averaged from t=0–2 in Part fear across tone 3, 4, and 5 (two-way RM ANOVA, main effect of tone: F(2, 34)=0.1031 ns, sex: F(1, 17)=12.05, p<0.01, tone × sex: F(2, 34)=7.572, p<0.01, Tukey post-hoc: Males tone 4 vs 5, p=0.0147, Females tone 4 vs 5, p=0.0423, Tone 3 Males vs Females, p=0.0061, Tone 5 Males and Females, p<0.0001). (**L**) Shock response in Part fear averaged from t=28–30 across all shock trials (two-way RM ANOVA, main effect of tone: F(2, 34)=0.7578 ns, sex: F(1, 17)=3.788 ns, p=0.07, tone × sex: F(2, 34)=0.3663 ns). (**M**) Shock omission response in Part fear averaged from t=28–30 across no shock trials (two-way RM ANOVA, main effect of tone: F(2, 34)=0.8610 ns, sex: F(1, 17)=2.054 ns, tone × sex: F(2, 34)=1.689 ns). (**N**) Fear recall traces in males. (**O**) Fear recall traces in females. (**P**) BNST Tone response averaged from t=0–2 for all tones in Full and Part fear recall (two-way ANOVA, main effect of fear: F(1, 30)=0.02371 ns, sex: F(1, 30)=0.001158 ns, fear × sex: F(1, 30)=0.01792 ns). (**Q**) Area under the curve (AUC) during tone presentations from t=0–30 (two-way ANOVA, main effect of fear: F(1, 30)=0.7822 ns, sex: F(1, 30)=7.444, p<0.05, fear × sex: F(1, 30)=2.036 ns). n=7–10/group. All data are shown as mean + SEM. *=p<0.05, **=p<0.01, ***=p<0.001, ****=p<0.0001.

The online version of this article includes the following source data and figure supplement(s) for figure 2:

*Figure 2 continued on next page*

*Figure 2 continued*

**Source data 1.** This source data contains all of the data from the graphs in this figure.

**Figure supplement 1.** Bed nucleus of the stria terminalis (BNST) activity negatively correlates with freezing.

much higher BNST tone response from Full fear males (*Figure 2F*). Interestingly, tone responses in males were also significantly higher than in females (*Figure 2F*). To minimize any differences driven by the number of tone presentations, we compared the responses to tone 3 in Full fear mice to the response to tone 4 in Part fear mice (*Figure 2G*). Here, we find no differences in tone response between Full and Part fear, however, reduced engagement of the BNST in females, regardless of fear type persists (*Figure 2G*).

Next, we quantified BNST responses to shock and found an interaction effect. Post-hoc tests reveal that BNST responses to shock were higher in Part fear males than Full, and that in the Full fear group, shock responses were higher in females than in males (*Figure 2I*). Comparing the final shock only, Part fear males show a greater response than Full fear males, while no difference in response amplitude is present in females (*Figure 2J*).

To characterize how BNST activity changes within subjects as uncertainty is introduced, we compared responses to tone onset in Part fear mice to tones 3, 4, and 5. Interestingly, BNST responses in males increase between tones 4 (preceded by shock omission) and 5 (preceded by shock) (*Figure 2K*), suggesting that the expected likelihood of receiving a shock may influence BNST activity. Furthermore, the opposite effect is present in females, with female BNST response decreasing between tones 4 and 5, suggesting sex differences in how the BNST processes uncertain threats. Responses to shock or shock omission did not differ across time or between sexes in part fear mice (*Figure 2L and M*). Interestingly, while females showed little BNST engagement to tone onset, BNST responses to shock and omission was similar to males (*Figure 2L–M*), indicating the sex differences exist primarily during the cue exposure in fear learning.

During fear recall, neither sex nor fear type affected BNST activity to tone onset (first 2 s of tone) (*Figure 2N–P*). However, we found that in males (*Figure 2N*), but not females (*Figure 2O*), BNST activity dipped below baseline during the full 30 sec tone presentations for both Full and Part fear (*Figure 2Q*). This suggests that sex differences to tone onset are only observed during fear conditioning and not recall, however, sex differences persist in BNST responses to the full tone exposure in recall. Overall, BNST dynamics were highly similar across groups and sexes during learning, and thus, the subtle differences in signaling may communicate the nuanced distinctions between Full and Part fear.

## CRF knockdown in the BNST potentiates partially reinforced fear acquisition and recall in females

CRF signaling in the BNST is critical for sustained fear (*Davis et al., 1997*; *Davis et al., 2010*), but classic studies do not distinguish local CRF signaling from CRF inputs to the BNST. To determine the role of CRF in BNST neurons in Part fear, we used a viral approach to genetically knock down CRF in the BNST (*Yu et al., 2021*). To do this, we injected an AAV encoding cre recombinase or GFP bilaterally into the BNST of Crh$^{lox/lox}$ mice (*Yu et al., 2021*) three weeks prior to behavior testing (*Figure 3A–C*). First, we compared freezing during fear conditioning and recall in male GFP and Cre mice. Part fear male mice regardless of virus froze more than Full fear mice when comparing all tone exposures (*Figure 3D*), but this effect did not appear when comparing just the final shocked tone exposure (*Figure 3E*). Like wild-type mice, male Part Fear mice froze more during tone 5 in fear conditioning than tones 3 and 4 (*Figure 3F*). There were no differences in freezing during baseline (*Figure 3G*) or first tone exposure (*Figure 3H*) in recall between male GFP and Cre mice, suggesting that CRF knockdown in the BNST does not alter full or part fear learning or recall in males.

In females, Part fear mice also freeze more than Full fear mice when comparing all tone exposures (*Figure 3I*), and interestingly, this effect persists when comparing just the last shocked tone (*Figure 3J*). Though CRF knockdown slightly increased freezing in Part fear only, this effect was not significant. However, comparisons of freezing across tones in just Part fear females reveal a main effect of CRF knockdown, where knockdown increases freezing across tones (*Figure 3K*). Interestingly, CRF knockdown selectively increased freezing in Part Fear mice during the baseline period of fear recall

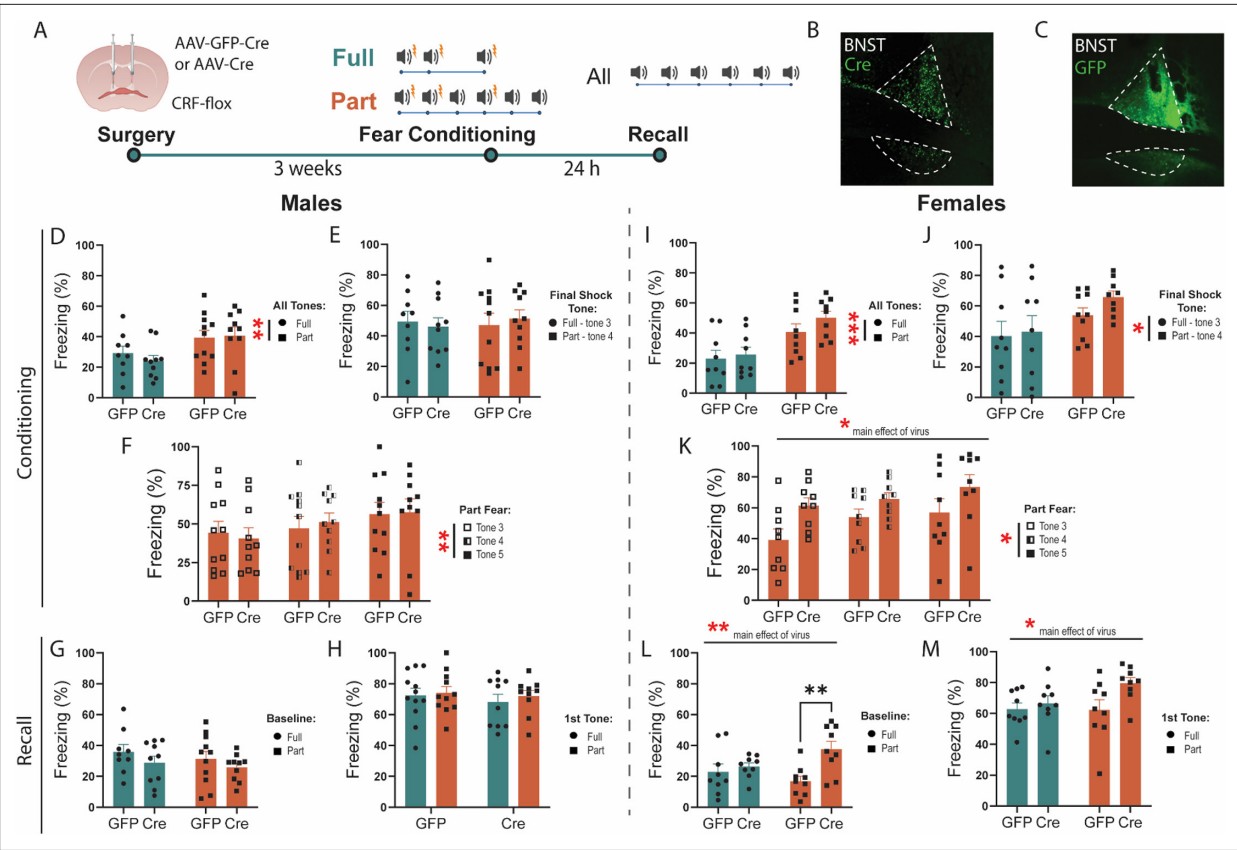

**Figure 3.** Corticotropin-releasing factor (CRF) knockdown in the bed nucleus of the stria terminalis (BNST) potentiates fear learning and recall in females. (**A**) Surgical schematic and timeline of CRF knockdown experiments. (**B, C**) Representative image of Cre and GFP virus expression in BNST. (**D**) Males percent freezing to all tone exposures during fear conditioning (two-way ANOVA, main effect of fear: F(1, 36)=7.871, p<0.01, virus: F(1, 36)=0.1492 ns, fear × virus: F(1, 36)=0.4452 ns). (**E**) Comparison of males Full tone 3 and Part tone 4 freezing response (two-way ANOVA, main effect of fear: F(1, 36)=0.04672 ns, virus: F(1, 36)=0.004876 ns, fear × virus: F(1, 36)=0.3099 ns). (**F**) Males freezing response in Part fear across tone 3, 4, and 5 (two-way RM ANOVA, main effect of tone: F(1.899, 36.08)=6.882, p<0.01, virus: F(1, 19)=0.0044 ns, tone × virus: F(2, 38)=0.5234 ns). (**G**) Freezing during baseline period in males Full and Part fear recall. (**H**) Freezing during first tone presentation in males Full and Part fear recall. (**I**) Females percent freezing to all tone exposures during fear conditioning (two-way ANOVA, main effect of fear: F(1, 32)=16.71, p<0.001, virus: F(1, 32)=1.393 ns, fear × virus: F(1, 32)=0.4367 ns). (**J**) Comparison of females Full tone 3 and Part tone 4 freezing response (two-way ANOVA, main effect of fear: F(1, 32)=5.265, p<0.05, virus: F(1, 32)=0.8753 ns, fear × virus: F(1, 32)=0.3144 ns). (**K**) Females freezing response in Part fear across tone 3, 4, and 5 (two-way RM ANOVA, main effect of tone: F(1.596, 25.54)=3.840, p<0.05, virus: F(1, 16)=5.768, p<0.05, tone × virus: F(2, 32)=0.4641 ns). (**L**) Freezing during baseline in females Full and Part fear recall (two-way ANOVA, main effect of fear: F(1, 32)=0.3949 ns, virus: F(1, 32)=8.591, p<0.01, fear × virus: F(1, 32)=4.283, p<0.05, Bonferroni's post-hoc: Part GFP vs Cre p=0.0025). (**M**) Freezing during first tone presentation in females Full and Part fear recall (two-way ANOVA, main effect of fear: F(1, 32)=1.628 ns, virus: F(1, 32)=4.545, p<0.05, fear × virus: F(1, 32)=1.886 ns). All data are shown as mean + SEM. *=p<0.05, **=p<0.01, ***=p<0.001, ****=p<0.0001.000.

The online version of this article includes the following source data for figure 3:

**Source data 1.** This source data contains all of the data from the graphs in this figure.

(*Figure 3L*). Furthermore, CRF knockdown also led to heightened freezing during recall first tone presentation regardless of fear type (*Figure 3M*). Overall, CRF knockdown led to an increase in fear learning in Part fear females (*Figure 3K*) and in fear recall in both Full and Part fear mice that appears to be driven by more pronounced changes in Part fear mice (*Figure 3L–M*). These results indicate that CRF knockdown in the BNST drives fear generalization and increased fear memory after partially reinforced fear in females, and that CRF differentially shapes passive coping in males and females.

## CRF knockdown in the BNST does not modulate anxiety-like or vigilance behaviors after fear

Next, we asked whether CRF knockdown alters anxiety-like and vigilance behaviors. Using the same approach described above, we knocked down CRF in the BNST and examined changes in anxiety-like behavior (*Figure 4A*). We found that CRF knockdown did not alter latency to feed in the NSF assay in male (*Figure 4B*) and female mice (*Figure 4C*), but both Full and Part fear conditioning led to a significant reduction in latency to feed compared to Ctrl mice in both sexes. Importantly, there was no effect of CRF knockdown on the amount of food consumed during the refeed portion of the test in any group, indicating that this approach does not alter consummatory behavior (*Figure 4D–E*). In the acoustic startle test, CRF knockdown did not affect startle amplitude in either sex (*Figure 4F–J*). Although there were no sex differences in startle amplitude in Ctrl mice, males had greater startle amplitude than females after both Full fear (*Figure 4G*) and Part fear (*Figure 4H*). To compare fear types, we separated males and females and focused on the loudest startle stimulus (120 dB) and found that fear conditioning potentiated startle amplitude in males (*Figure 4I*) but not females (*Figure 4J*). This is like what was found in *Figure 1*, except that male Part Fear mice did not exhibit a robust increase. Overall, these findings suggest that CRF knockdown in the BNST does not shape these avoidance or arousal behaviors.

## Activity of BNST<sup>CRF</sup> neurons during fear

We previously found that BNST<sup>CRF</sup> neurons contribute to fear encoding (*Marcinkiewcz et al., 2016*) but only when serotonin levels are increased in the BNST. This suggests that in some situations, BNST<sup>CRF</sup> can directly regulate fear learning. Given the effects of CRF knockdown on fear learning and recall in female mice (*Figure 3*), we hypothesized that males and females had differential engagement of BNST<sup>CRF</sup> during fear conditioning and recall, which could drive the behavioral consequences of fear learning. To record BNST<sup>CRF</sup> activity, we injected an AAV encoding a cre-dependent GCaMP8m into the BNST of *Crh-cre* mice and implanted GRIN lenses for 1-photon calcium imaging (*Figure 5A*).

First, we compared BNST<sup>CRF</sup> activity between Full and Part fear conditioning and recall. During fear conditioning, we observed that roughly a third of cells were excited by shock in both sexes, and the rest were inhibited (*Figure 5B–C*). The BNST is sexually dimorphic, and females have more BNST<sup>CRF</sup> neurons than males (*Salvatore et al., 2018*; *Chudoba and Dabrowska, 2023*), therefore, we first examined if there were any baseline differences between male and female BNST<sup>CRF</sup> activity. We compared BNST<sup>CRF</sup> activity during the baseline period in fear conditioning, before any shock or tone exposures, and found that female BNST<sup>CRF</sup> neurons were more active than males (*Figure 5D*). Interestingly, activity decreased throughout the conditioning session for both sexes, but the reduction was more pronounced in females, as revealed by comparison of activity during the consolidation period of fear conditioning (last 2 min of trial) to baseline (*Figure 5E*). When comparing BNST<sup>CRF</sup> event frequency during tone exposures, we found that females, regardless of fear group had higher activity than males (*Figure 5F*). This could be due to females having higher activity overall in the BNST<sup>CRF</sup> neurons. In fact, when comparing the last tone shock between Part and Full fear, we find no significant effects of sex or fear conditioning type (*Figure 5G*). To characterize BNST<sup>CRF</sup> responses to shock, we compared the 10 s following the final shock and found no differences in event frequency in BNST<sup>CRF</sup> between Full or Part fear or between sexes (*Figure 5H*).

Differences in expression of fear during recall between fear groups and sexes point to sex-specific roles of BNST<sup>CRF</sup> neurons during fear recall. Interestingly, there were no sex differences in event frequency during the baseline period in recall (*Figure 5J*), but similar to conditioning, BNST<sup>CRF</sup> activity decreased significantly between baseline and session end during recall (*Figure 5K*). However, Part fear male BNST<sup>CRF</sup> activity showed a smaller reduction compared to Full fear males (*Figure 5I & K*), suggesting that male BNST<sup>CRF</sup> neurons are more engaged during recall of uncertain fear. This remains true when looking at the first and last tone exposure, where Part fear males showed an increase in Event frequency from first to last tone compared to Full fear males and females (*Figure 5M*). Furthermore, elevated engagement of BNST<sup>CRF</sup> neurons was also seen in Part fear mice, regardless of sex, during tone exposure compared to Full fear mice (*Figure 5L*). Interestingly, comparisons of activity during the 10 s after tone end (when mice might expect to receive a shock) reveal that males show a reduction in activity between first and last tone regardless of fear group, while females show an increase in BNST<sup>CRF</sup> activity (*Figure 5N*).

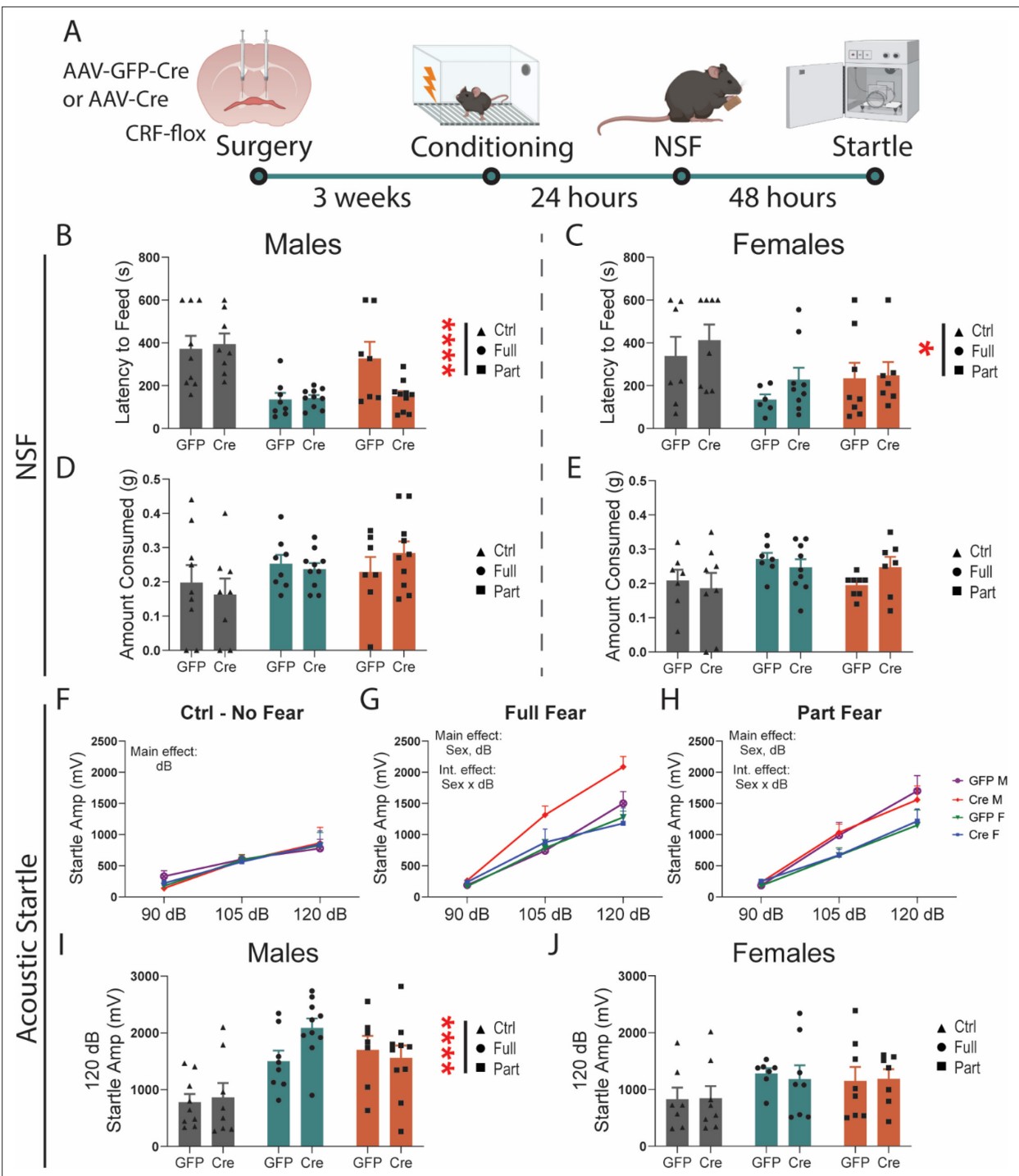

**Figure 4.** Corticotropin-releasing factor (CRF) knockdown in the bed nucleus of the stria terminalis (BNST) does not modulate anxiety-like or vigilance behaviors after fear. (**A**) Surgical schematic and behavioral timeline. (**B**) Latency to feed in the novelty-induced suppression of feeding (NSF) test in males (two-way ANOVA, main effect of fear: F(2, 46)=15.24, p<0.0001, main effect of virus: F(1, 46)=1.808 ns, fear × virus interaction F(2, 46)=2.972 ns, *p*=0.06). (**C**) Latency to feed in the novelty-induced suppression of feeding (NSF) test in females (two-way ANOVA, main effect of fear F(2, 39)=1.190, p<0.05, main effect of virus F(1, 39)=1.190 ns, fear × virus interaction F(2, 39)=0.1847 ns). (**D**) Amount consumed during refeed test in males. (**E**) Amount consumed during refeed test in females. (**F**) Acoustic startle response in Ctrl mice (three-way RM ANOVA, main effect of dB: F(1.101, 30.83)=27.40, p<0.0001, virus: F(1, 28)=.0298 ns, sex: F(1, 28)=0.0167 ns). (**G**) Acoustic startle response in Full fear mice (three-way RM ANOVA, main effect of dB: F(1.701, 49.33)=126.3, p<0.0001, virus: F(1, 29)=3.590 ns, *p*=0.07, sex: F(1, 29)=5.260, p0.05, dB × Sex F(2, 58)=5.717, p<0.01). (**H**) Acoustic startle response in Part fear mice (three-way RM ANOVA, main effect of dB: F(1.466, 41.05)=88.49, p<0.0001, virus: F(1, 28)=.0037 ns, sex: F(1, 28)=4.763, p<0.05, dB × Sex F(2, 56)=3.965, p<0.05). (**I**) Acoustic startle response at 120 dB in males (two-way ANOVA, main effect of main effect of

*Figure 4 continued on next page*

*Figure 4 continued*

fear: F(2, 46) = 12.96 p<0.0001, main effect of virus: F(1, 46)=1.131 ns, fear × virus interaction: F(2, 46)=1.666 ns). (J) Acoustic startle response at 120 dB in females (two-way ANOVA, main effect of main effect of fear: F(2, 39) = 2.046 ns, main effect of virus: F(1, 39)=0.0080 ns, fear × virus interaction: F(2, 39)=0.0592 ns). N=7–10/group. All data are shown as mean ± SEM. *=p<0.05, **=p<0.01, ***=p<0.001, ****=p<0.0001.

The online version of this article includes the following source data for figure 4:

**Source data 1.** This source data contains all of the data from the graphs in this figure.

Furthermore, we noted sex differences in Part fear conditioning (*Figure 6*). BNST$^{CRF}$ event frequency during tones 3, 4, and 5 differed between sexes; while male mice showed similar levels of BNST event frequency during tones, BNST$^{CRF}$ event frequency decreased across tone exposures in females (*Figure 6E*). This effect is similar to what we saw with general BNST activity in fiber photometry recordings (*Figure 2*) and suggests that different neural processes underlie sustained fear encoding in females. Half of the tones in Part fear conditioning terminate in shock while the other half do not (shock omission), thus we compared event frequency in the 10 s window following a shock or omission trial (*Figure 6F–I*). Both males and females demonstrated a slight increase in activity during the second shock (*Figure 6F*). Interestingly, while males had some BNST$^{CRF}$ activity after shock omission, female mice showed little engagement of BNST$^{CRF}$ neurons following each omission trial (*Figure 6G*). To compare BNST$^{CRF}$ activity post-shock or omission, we examined only the final shock and omission trials (*Figure 6H*). Male BNST$^{CRF}$ neurons are more engaged during omission than shock, while female BNST is more engaged during shock than omission (*Figure 6H–I*). Overall, these data demonstrate sex differences in BNST$^{CRF}$ activity during fear conditioning, where females show greater silencing of BNST$^{CRF}$ neurons compared to males. This may also indicate sex differences in how BNST$^{CRF}$ encodes negative prediction error during Part fear conditioning.

## Discussion

### Partially reinforced fear conditioning drives sustained fear in males

Phasic fear refers to defensive responses triggered by clear and imminent threats, which dissipate rapidly once the threat is removed (*Davis et al., 2010*). In contrast, sustained fear is characterized by a prolonged state of apprehension, driven by distant or ambiguous cues that persist over time. Full fear and Part fear conditioning are rodent experimental paradigms designed to model phasic and sustained fear, respectively. Both consist of three tone-shock pairings, but Part fear includes an additional three tones that do not co-terminate with a shock (*Figure 1A*). This small difference in the Part fear paradigm does not consistently drive distinct behavioral effects in mice. Although we observed differences in freezing when comparing all tone exposures, these are likely due to Part fear consisting of three more tones, as there were no significant differences between Full fear and Part fear groups during specific tone exposures in fear conditioning and recall.

The Part Fear paradigm allows for careful dissection of uncertain fear learning by allowing for within-mouse comparisons before and after uncertainty are introduced. A notable and consistent finding was an increase in freezing behavior during tone 5 presentation in the Part fear group compared to tones 3 and 4 (*Figures 1 and 3*). This pattern offers insights into sustained fear encoding, as it highlights how uncertainty can shape defensive behaviors. As tones 1 and 2 co-terminate with shock, mice are likely to perceive the third tone as a threat cue. Tone 4 introduces uncertainty, as it follows a tone that did not co-terminate with a shock. By tone 5, the ambiguity is heightened, as mice have now experienced both tone-shock pairings and tone-shock omissions, making tone 5 the first truly ambiguous tone in the Part fear paradigm. This escalation in ambiguity likely underpins the observed increase in freezing and provides a valuable model for studying sustained fear.

Our findings demonstrate that fear conditioning increases the acoustic startle response in males only (*Figures 1 and 4*). In wild-type mice, this heightened startle response is driven by a pronounced increase in hypervigilance in males who underwent Part fear conditioning (*Figure 1R*). Notably, this effect was observed three days after fear learning, indicating a sustained increase in arousal. The acoustic startle test serves as a valuable measure of psychomotor arousal, and elevated startle responses are commonly seen in individuals with anxiety disorders (*Johnson et al., 2012*; *Gewirtz et al., 1998*). Hypervigilance and heightened threat responsiveness are also core symptoms of

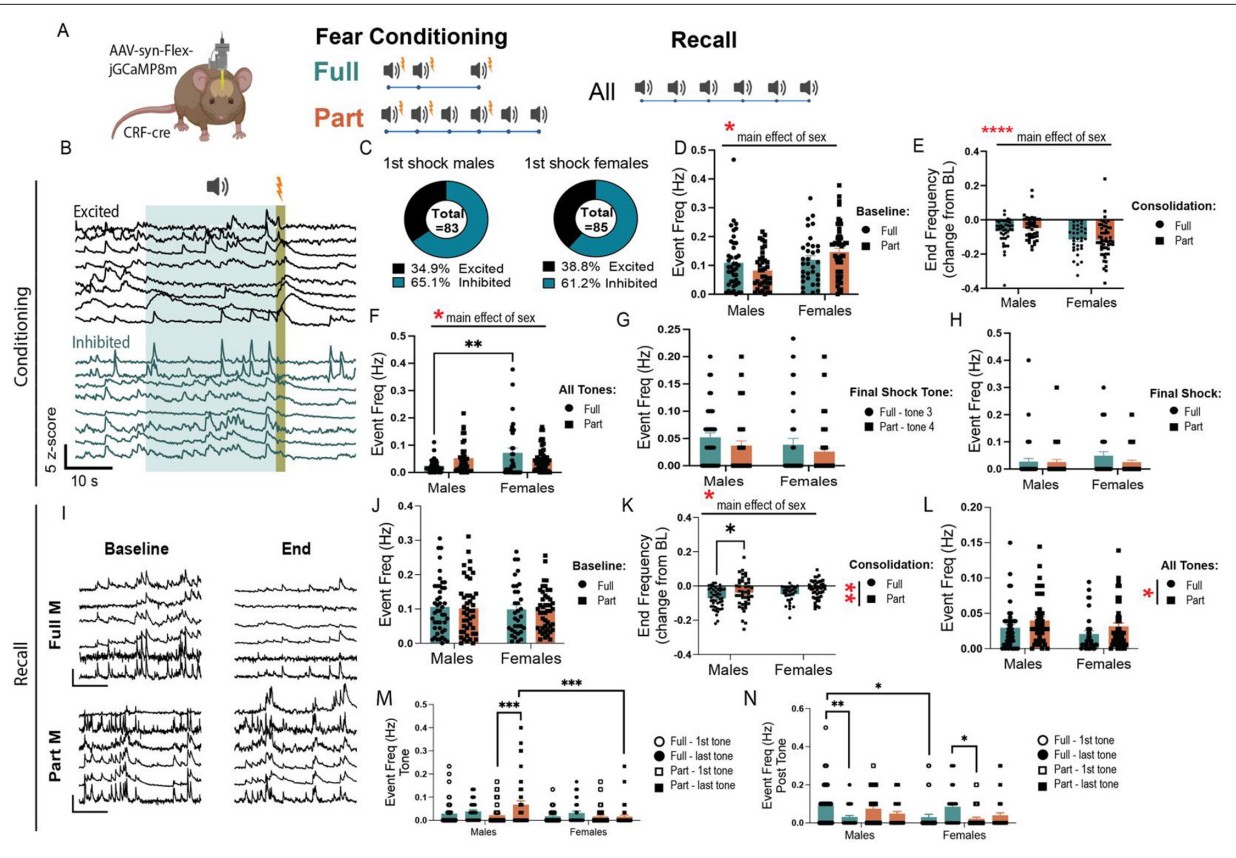

**Figure 5.** Corticotropin-releasing factor (CRF) neurons differentially encode fear recall in males and females. (**A**) Viral strategy and experimental design schematic. (**B**) Example traces of different responses to foot shock. (**C**) Pie charts showing the proportion of cells excited and inhibited in response to the first foot shock in males and females, fear groups combined. (**D**) Event frequency during baseline period in fear conditioning baseline (two-way ANOVA, main effect of fear: F(1, 161)=0.00030 ns, sex: F(1, 161)=7.558, p<0.01, fear × sex: F(1, 161)=3.987, p<0.05). (**E**) Event frequency as a change from baseline during consolidation period at the end of fear conditioning (two-way ANOVA, main effect of fear: F(1, 161)=0.3744 ns, sex: F(1, 161)=18.91, p<0.0001, fear × sex: F(1, 161)=0.6552 ns). (**F**) Event frequency during all tone presentations in fear conditioning (two-way ANOVA, main effect of fear: F(1, 161)=0.1263 ns, sex: F(1, 161)=6.462, p<0.05, fear × sex: F(1, 161)=7.135, p<0.01, Bonferroni Post Hoc: Full, Males vs Females, p=0.0011). (**G**) Event frequency during final shock tone, Full fear tone 3 and Part fear tone 4 (two-way ANOVA, main effect of fear: F(1, 161)=0.2.638 ns, sex: F(1, 161)=2.060 ns, fear × sex: F(1, 161)=0.02164 ns). (**H**) final shock in Full and Part fear conditioning (two-way ANOVA, main effect of fear: F(1, 161)=1.377 ns, sex: F(1, 161)=0.9991 ns, fear × sex: F(1, 161)=1.125 ns). (**I**) Example traces during baseline and trial end in fear recall. (**J**) Full and Part fear recall baseline period event frequency (two-way ANOVA, main effect of fear: F(1, 162)=0.05488 ns, sex: F(1, 162)=0.2847 ns, fear × sex: F(1, 162)=0.0058 ns). (**K**) Full and Part fear recall trial end event frequency as a change from baseline (two-way ANOVA, main effect of fear: F(1, 162)=8.357, p<0.01, sex: F(1, 162)=4.143, p<0.05, fear × sex: F(1, 162)=0.1101 ns, Bonferroni's post-hoc: Males Full vs Part, p=0.0384). (**L**) Event frequency during all tone presentations in recall (two-way ANOVA, main effect of fear: F(1, 162)=4.593, p<0.05, sex: F(1, 162)=3.202 ns, p=0.08, fear × sex: F(1, 162)=0.0013 ns). (**M**) Comparison of event frequency during first and last tone presentation in recall (three-way ANOVA, main effect of fear: F(1, 162)=10.85, p<0.01, sex: F(1, 162)=7.385, p<0.01, fear × sex: F(1, 162)=3.981, p<0.05, fear × sex × tone: F(1, 162)=5.916, p<0.05, Tukey's post hoc: Males Part first tone vs Males Part last tone, p=0.0004, Males Part last tone vs Females Part last tone, p=0.0005). (**N**) Comparison of event frequency during the 10 s following the first and last tone presentation in recall (three-way ANOVA, main effect of sex: F(1, 162)=3.340 ns, p=0.07, fear × sex: F(1, 162)=20.62, p<0.0001, fear × sex × tone: F(1, 162)=4.334, p<0.05, Tukey's post hoc: Males Full first tone vs Males Full last tone, p=0.0069, Males Full first tone vs Females Full first tone, p=0.0237, Females Full last tone vs Females Part first tone, p=.0253). For fear conditioning: Full Males: n=41 cells from 6 mice. Part Males: n=39 cells from 5 mice. Full females: n=33 cells from 6 mice. Part females: n=52 cells from 5 mice. For fear recall: Full Males: n=45 cells from 6 mice. Part Males: n=44 cells from 5 mice. Full females: n=33 cells from 6 mice. Part females: n=45 cells from 5 mice. All data are shown as mean + SEM. *=p<0.05, **=p<0.01, ***=p<0.001, ****=p<0.0001.

The online version of this article includes the following source data for figure 5:

**Source data 1.** This source data contains all of the data from the graphs in this figure.

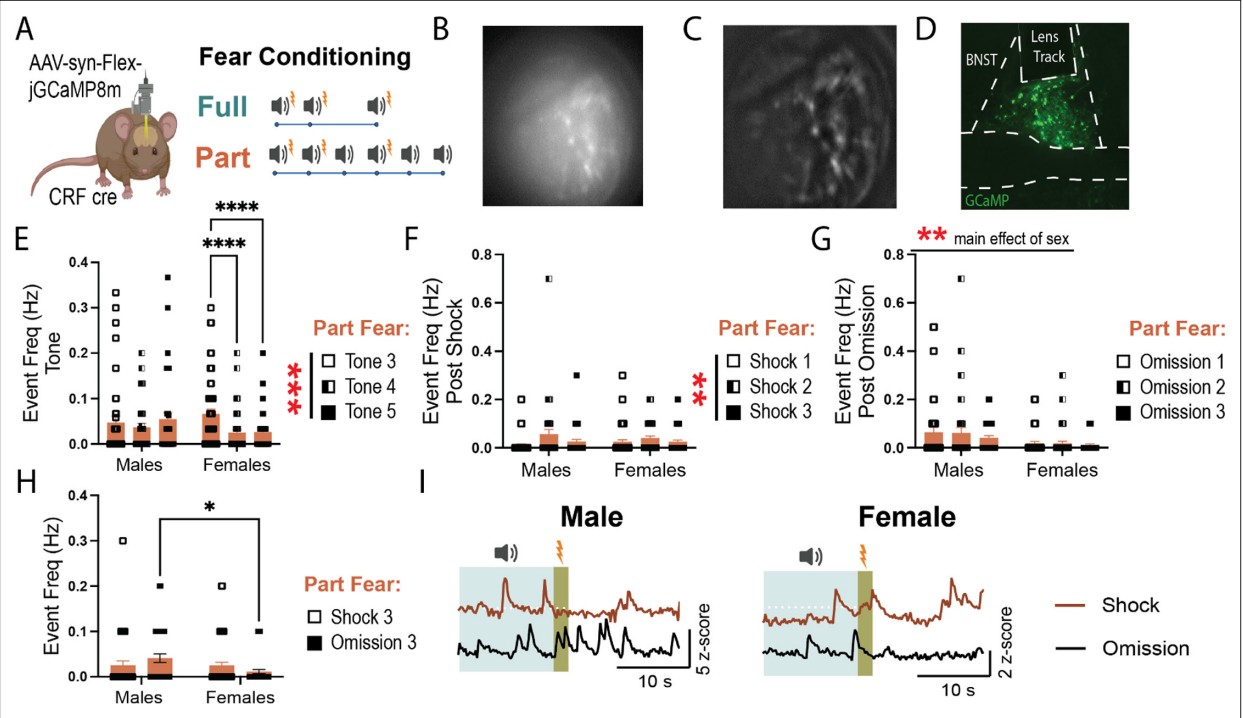

**Figure 6.** Sex differences in BNST[CRF] neuron encoding of Part fear conditioning. (**A**) Experimental design schematic for 1-photon recordings. (**B**) Representative raw FOV. (**C**) Representative processed FOV. (**D**) Representative image showing GCaMP expression and GRIN lens placement. (**E**) Event frequency during partial fear tone 3, 4, and 5 presentations (two-way RM ANOVA, main effect of tone: $F_{(2, 180)}=7.330$, $p<0.001$, sex: $F_{(1, 90)}=0.3467$ ns, tone × sex: $F_{(2, 180)}=6.365$, $p<0.01$, Bonferroni's post-hoc: Females tone 3 vs 4, $p<0.0001$, Females tone 3 vs 5, $p<0.0001$). (**F**) Event frequency during the 10 s post-shock across all shock trials (two-way RM ANOVA, main effect of shock: $F_{(2, 178)}=5.386$, $p<0.01$, sex: $F_{(1, 89)}=0.0044$ ns, shock × sex: $F_{(2, 178)}=1.243$ ns). (**G**) Event frequency during the 10 s post-omission across all omission trials (two-way RM ANOVA, main effect of omission: $F_{(2, 178)}=1.412$ ns, sex: $F_{(1, 89)}=8.552$, $p<0.01$, omission × sex: $F_{(2, 178)}=0.3291$ ns). (**H**) Event frequency during the 10 s final post-shock or post-omission window during partial fear (two-way RM ANOVA, main effect of shock/omission: $F_{(1, 89)}=0.0209$ ns, sex: $F_{(1, 89)}=3.174$ ns $p=0.08$, shock/omission × sex: $F_{(1, 89)}=4.697$, $p<0.05$, Bonferroni's post-hoc: Omission Males vs Females, $p=0.0135$). (**I**) Representative traces showing a response to shock and shock-omission trials for partial fear only in males and females. All data are shown as mean + SEM. *=$p<0.05$, **=$p<0.01$, ***=$p<0.001$, ****=$p<0.0001$.

The online version of this article includes the following source data for figure 6:

**Source data 1.** This source data contains all of the data from the graphs in this figure.

post-traumatic stress disorder (PTSD) in humans (*Johnson et al., 2012*; *Fragkaki et al., 2017*; *Ressler et al., 2022*). Interestingly, a history of fear conditioning did not lead to any significant behavioral differences in females, either in the NSF (novelty-suppressed feeding) test or in startle amplitude. This highlights a significant gap in the literature, as studies exploring the effects of fear exposure in females remain limited. Further research is essential to understand the factors driving behavioral responses in females after fear conditioning, especially since NSF and startle are not the only tests measuring aspects of anxiety and vigilance. Exploring additional behavioral paradigms will be critical to uncovering the nuances of fear responses in females.

## BNST dynamics during fear

While classical fear literature emphasizes the amygdala's critical role in phasic fear and the BNST's involvement in sustained fear, recent findings suggest that the BNST contributes to both fear types but is more prominently recruited and indispensable (*Goode et al., 2019*) for sustained fear. For example, a recent neuroimaging study in humans found that the BNST showed greater engagement than the amygdala during ambiguous threats, highlighting its role in processing sustained fear across species (*Naaz et al., 2019*). Accordingly, we investigated the role of BNST neurons in both phasic and sustained fear in male and female mice using fiber photometry to measure BNST population activity (*Figure 2*). Our data support the hypothesis that the BNST is more responsive to uncertain fear cues in males, but not females. This finding is critical, as few studies have examined sex differences in BNST

responses, despite the higher prevalence of anxiety and mood disorders related to sustained fear, such as PTSD, in women.

In male mice, BNST activity was significantly higher in response to tone 5 compared to tones 3 and 4 during Part fear condition (*Figure 2K*), and BNST tone responses were highest in Part fear males (*Figure 2F*). This may represent that the BNST encodes the expected likelihood of a negative outcome. This is consistent with in vivo electrophysiology recordings suggesting that roughly 40% of BNST neurons encode attention or uncertainty of aversive contingency (*Bjorni et al., 2020*). Furthermore, human fMRI studies have linked BNST activity to anticipation of threat, particularly those that are ambiguous. We also found that male Part fear mice exhibited the highest BNST response to the final shock compared to Full fear males and females (*Figure 2J*). These findings indicate that BNST activity is particularly heightened in males during sustained fear and matches the increased vigilance observed behaviorally (*Figure 1*). In contrast, female mice showed minimal BNST responses to tone presentations during fear conditioning (*Figure 2F–H & K*). Notably, in Part fear, females exhibited a reduction, rather than the expected increase, in BNST activity in response to tone 5 (*Figure 2K*). Importantly, these sex differences were specific to tone presentation, as BNST activity did not differ between sexes during shock (*Figure 2L*) or shock omission (*Figure 2M*) in Part fear.

Sex differences in BNST dynamics also emerged during fear recall, where we showed that BNST activity was overall suppressed during the full 30 s tones in males but not in females, with a slightly greater suppression in Part fear males versus Full fear (*Figure 2Q*). This is consistent with findings that an inhibitory input to the BNST from the medial prefrontal cortex is engaged during Part fear recall (*Glover et al., 2020*). Additionally, during both conditioning and recall, activity in the BNST negatively correlated with freezing (*Figure 2—figure supplement 1*), suggesting that this is an innate property of the BNST and is not altered by fear conditioning. These results are consistent with findings that the BNST mediates the expression of proactive avoidance behaviors to prevent harm (*Guerra et al., 2022*). Interestingly, the inhibition of a specific BLA-BNST pathway reduces fear conditioning to a sustained cue in males but not females, suggesting that different BNST subpopulations may mediate sex-specific responses (*Vantrease et al., 2022*). These results emphasize the importance of studying BNST microcircuits and their contributions to fear processing across sexes.

## CRF knockdown alters fear learning and expression in females only

The BNST and CRF signaling are critical for sustained fear across species (*Davis et al., 2010*; *Walker et al., 2009*; *Bangasser and Wiersielis, 2018*; *Salvatore et al., 2018*; *Somerville et al., 2010*). However, the mechanisms through which CRF in the BNST contributes to fear expression remain unclear. The BNST contains CRF-expressing neurons that release CRF locally and to other brain regions, and receives CRF inputs from external sources (*Marcinkiewcz et al., 2016*; *Pati et al., 2020*). These distinct CRF nodes may differentially influence fear expression. Studies examining sustained fear in rodents point towards distinct roles for CRF inputs to the BNST and BNST$^{CRF}$ neurons. CRF infusions into the BNST increase fear expression (*Walker et al., 2009*), while CRF receptor antagonism reduces sustained fear expression (*Risbrough and Stein, 2006*), suggesting that CRF inputs to the BNST are crucial for sustained fear expression. Conversely, one study using viral overexpression of CRF in the BNST of males found that it weakened sustained fear expression (*Sink et al., 2013*), which suggests that CRF inputs to the BNST may drive sustained fear, while exaggerated BNST$^{CRF}$ signaling may attenuate sustained fear. These studies have been done in males only; thus we focused on understanding how CRF knockdown in BNST$^{CRF}$ neurons affects behavior in males and females using a genetic knockdown approach (*Figures 3–4*).

Since CRF overexpression weakened sustained fear in male rats, we hypothesized that CRF knockdown would enhance sustained fear in male mice, however, we found that CRF knockdown in the BNST led to no changes in Full or Part fear conditioning or recall in males (*Figure 3D–H*). This may be due to a species difference or a difference in the sustained fear paradigm used. The partially reinforced cued fear paradigm used in these studies may not be salient or different enough from a fully reinforced fear paradigm to reliably elicit behavioral differences and future studies using more prolonged sustained fear paradigms may be more informative. This is supported by the evidence that BNST$^{CRF}$ signaling is not necessary for phasic cued fear conditioning, as chemogenetic inhibition of male BNST$^{CRF}$ neurons during conditioning or consolidation of cued fear conditioning has no effect on fear (*Marcinkiewcz et al., 2016*; *Bruzsik et al., 2021*). Interestingly, CRF knockdown in females

led to a significant increase in freezing behavior across tones in Part fear conditioning (*Figure 3K*) and an increase in freezing during baseline and the first tone presentation during recall in both Part and Full fear (*Figure 3L–M*). These effects appear driven by Part fear females, as post-hoc tests show that knockdown led to generalized fear expression in Part fear female mice compared to GFP controls (*Figure 3L*). This suggests that CRF knockdown only impacts fear in female mice, driving a general increase in fear learning and expression.

Intra-BNST infusion of CRF potentiates acoustic startle response (*Lee and Davis, 1997*), and locally projecting BNST CRF neurons drive anxiety by inhibiting anxiolytic outputs from BNST (*Marcinkiewcz et al., 2016*; *Pati et al., 2020*). Thus, we hypothesized that knockdown of CRF in the BNST would block the increased acoustic startle we observed in males after fear (*Figure 1N–R*). However, CRF knockdown had no effect in the acoustic startle test or the NSF assay. CRF inputs to the BNST are also important for fear-related behaviors. It may be that behavioral effects in the startle test or NSF are instead driven by CRF inputs to the BNST, thus, future experiments knocking down CRF R1 and R2 will clarify these effects. BNST$^{CRF}$ neurons project both locally and out of the BNST (*Marcinkiewcz et al., 2016*; *Pati et al., 2020*), which may indicate that CRF signaling within the BNST versus in target regions of BNST$^{CRF}$ differentially alters acoustic startle reflex. Alternatively, the lack of an effect of CRF knockdown on acoustic startle after fear suggests a potential CRF-independent role of BNST$^{CRF}$ neurons instead, as this population also releases GABA (*Dabrowska et al., 2011*). This is especially interesting, as we recently found that knockdown of GABA vs CRF BNST$^{CRF}$ neurons led to differential regulation of operant ethanol self-administration (*Gianessi et al., 2023*). Overall, these findings are consistent with reports that CRF plays divergent roles in anxiety-like and stress behaviors in males and females (*Bangasser and Wiersielis, 2018*; *Salvatore et al., 2018*; *Bangasser and Valentino, 2012*).

## BNST$^{CRF}$ neurons are persistently active during Part fear expression in males

Given the established link between CRF levels in the BNST and sustained fear, we hypothesized that BNST$^{CRF}$ neuron activity would be elevated after Part, but not Full fear conditioning. Unexpectedly, we found minimal differences in BNST$^{CRF}$ spiking between fear groups in 1-photon calcium imaging experiments. However, these results align with our previous findings showing that chemogenetic inhibition of CRF neurons does not affect full fear responses (*Marcinkiewcz et al., 2016*), and suggest that CRF neuron activity during fear conditioning does not play a crucial role in the distinction between phasic and sustained fear.

Interestingly, male Part fear mice demonstrated persistent engagement of BNST$^{CRF}$ neurons during the recall baseline through the end of the session (*Figure 5I*). They also exhibited increased engagement during the last tone compared to the first tone in recall (*Figure 5M*). Persistent BNST$^{CRF}$ activity is particularly interesting, as elevated central CRF levels and CRF signaling hyperactivity have been documented in people with stress-related disorders (*Henckens et al., 2016*). Accordingly, elevated BNST CRF neuron activity may drive the behavioral alterations observed following Part fear, like the hypervigilance observed in *Figure 1*.

Consistent with other studies (*Levine et al., 2021*), we found that females overall had higher basal BNST$^{CRF}$ tone than males (*Figure 5D*). In *Figure 6*, we observed that female BNST$^{CRF}$ spiking decreased during tones 4 and 5 in Part fear (*Figure 6E*). Additionally, while male BNST$^{CRF}$ neurons exhibited higher event frequency following shock omission than after shock trials, female BNST$^{CRF}$ activity was suppressed after omission trials (*Figure 6F–I*), suggesting reduced engagement during ambiguity in females. These findings suggest nuanced sex- and condition-dependent roles of BNST$^{CRF}$ neurons in fear processing.

Notably, we did not observe major sex differences in BNST$^{CRF}$ activity during fear recall despite robust behavioral differences after CRF knockdown. There are a few possible explanations for this. First, Ca2+ activity in CRF cell bodies may not necessarily reflect the release of CRF itself. Others have shown that regulation of dopamine terminals leads to changes in dopamine release that do not reflect cell body activity (*Kramer et al., 2020*); thus, it is possible that there are sex differences in CRF release that we cannot capture using this method. Second, CRF signaling in the BNST drives distinct behavioral and cellular responses to stress in male and female rats (*Chudoba and Dabrowska, 2023*; *Uchida et al., 2019*; *Martianova et al., 2019*). Thus, sex differences in behavior may emerge downstream of BNST$^{CRF}$ neuron activity.

Furthermore, it is important to note that BNST[CRF] neurons comprise two dissociable populations with opposing roles in fear processing. CRF interneurons projecting within the BNST inhibit anxiolytic and fear-dampening outputs to areas such as the ventral tegmental area and lateral hypothalamus, thereby driving fear and anxiety (*Kavaliers and Choleris, 2001*; *Pati et al., 2020*). Consequently, recording activity from all BNST[CRF] neurons may obscure critical changes occurring within specific subpopulations. Further studies aimed at disentangling the contributions of these distinct populations during partially reinforced fear are needed.

## Conclusions

Previous studies have not extensively explored sex differences in neural responses to uncertain threat, and our findings reveal critical sex differences in BNST function and behavior during sustained fear. Males exhibited hypervigilance and heightened BNST engagement with increasing ambiguity, while females showed reduced BNST engagement. Additionally, knockdown of CRF in BNST[CRF] neurons led to increased fear expression in female mice only, suggesting that CRF in the BNST is important for the normal expression of fear in females. The consistent blunting of BNST population activity (*Figure 2*) and single cell BNST[CRF] activity (*Figures 5–6*) during tone exposures in Part fear females are particularly compelling, as women are more likely to develop PTSD, a condition associated with impaired discrimination between threat and safety cues (*Ressler et al., 2022*). These findings suggest potential innate sex differences in BNST-mediated threat cue discrimination, which may drive sex-specific vulnerabilities to affective mood disorders. These differences provide insight into the neural circuits underlying fear and stress disorders, particularly in females, who face a higher prevalence of anxiety and mood disorders. Future research focusing on BNST microcircuits and sex-specific adaptations is essential to develop targeted interventions.

# Materials and methods

## Mice

All experiments were conducted in accordance with the University of North Carolina at Chapel Hill's Institutional Animal Care and Use Committee's guidelines under protocols 19–078 and 22–015. All mice were group-housed and maintained on a 12 hr/12 hr light-dark cycle with lights on at 7 AM and had access to standard rodent chow and water ad libitum. Male and Female C57BL/6J mice>8 weeks of age were purchased from Jackson Laboratories (Stock number 000664, Bar Harbor, ME). *Crh*-cre (Stock number 012704), and *Crh*[lox/lox] mice were bred in-house and maintained on a C57BL/6J background.

## Surgery

Mice were anesthetized using isoflurane (4% induction, 1–2% maintenance). Mice were administered buprenorphine (0.1 mg/kg, s.c.) during surgery and were also given acetaminophen (80 mg/200 mL in drinking water) 1 day before and for at least 3 days following surgery. AAV constructs were delivered to target regions using Hamilton microsyringes at the following coordinates; dBNST: AP +0.3 mm, ML +/-0.9 mm, DV –4.35. vBNST: AP +0.3 mm, ML +/-0.9 mm, DV –4.6. For fiber photometry experiments, fiber optic cannulae (FOC) were cut to length and implanted at the same coordinates and cemented to the skull with dental cement (C&B Metabond, Parkell). For miniature microscope experiments, GRIN lenses (0.6 × 7.3 mm) with integrated baseplates (Inscopix, Mountain View, CA) were implanted 0.2 mm above the target coordinates and lowered at a rate of 0.2 mm/min. Mice recovered for >3 weeks prior to the start of experiments.

## Viruses

For CRF deletion experiments, AAV5-CaMKiia-cre-GFP and AAV5-CaMKiia-GFP were obtained from the UNC Vector Core (Chapel Hill, NC). Viruses for fiber photometry and 1-photon calcium imaging detailed below.

## Behavior

### Fear conditioning

Fear conditioning was conducted using a 3-day protocol. On the first day (habituation), mice were placed into a fear conditioning chamber (context A, Med Associates, St. Albans, VT) equipped with a shock grid floor and clear plexiglass walls and allowed to forage for 5 min. Context A was cleaned with a 20% ethanol +1% vanilla extract solution to provide a scent cue. On day 2 (fear learning), mice were returned to context A. Following a 3 min baseline, mice were presented with tone-shock pairings (tone: 30 s, 80 dB, 3 KHz, shock: 0.6 mA, 2 s), or unpaired tones depending on the experimental group. Mice subjected to fully reinforced fear (Full) received three tone-shock pairings separated by inter-trial intervals of 60 and 90 s. Mice undergoing partially reinforced fear (Part) received three tone-shock pairings as well as three unpaired tones interspersed such that tones 1, 2, and 4 ended in shock, and 3, 5, and 6 did not. Inter-trial intervals were as follows: 60-20-40-20-20. Control mice (Ctrl) received six tones at the same timing as the Part group, except the mice did not receive foot shocks. Following tone-shock presentations, mice remained in context A for a 2 min consolidation period (End). On the third day, mice were placed into a novel context with white plastic flooring, curved walls, and cleaned with 0.5% acetic acid. Following a 3 min baseline, six tones (identical to fear learning) were presented, separated by a random ITI of 20–60 s. Behavior hardware was controlled by Ethovision XT software (Noldus Inc).

### NSF

Mice were food-deprived overnight prior to the start of NSF. On test day, a single pellet of standard chow was placed into the center of a brightly lit (1350 lux) white Plexiglas open field chamber. The floor was covered with normal cage bedding. Mice were placed into the corner of the chamber. The trial was ended when the mouse ate for a period of 3 s or more. If mice did not eat, the trial was ended after 10 min and mice were assigned a maximum latency of 600 s. After the test, mice were returned to a dimly lit clean cage and allowed to eat for 10 min, and the amount consumed was recorded.

### Acoustic startle

Mice were semi-restrained in Plexiglas tubes mounted to a platform with an accelerometer attached and placed inside a sound-attenuating chamber (all hardware and software from San Diego Instruments, San Diego, CA). This context was separate from fear conditioning chambers. Following a 5 min acclimation period, white noise bursts of varying intensity were played, and startle responses were measured. Four different stimulus intensities were tested: 90 dB, 105 dB, 120 dB, and 0 dB (control) separated by a random ITI of 30–60 s. Each session consisted of 10 blocks of four trials, one of each intensity, such that there were 10 total trials at each stimulus intensity. Startle response peaks were detected automatically using Startle Response Lab software.

## Fiber photometry

GCaMP6s (AAV5-hsyn-GCaMP6s, UPenn Vector Core, Philadelphia, PA) was injected into the BNST of C57BL/6J mice and fibers were implanted unilaterally in the BNST. Fiber optic cannulae (200 μm, 0.37nA, Neurophotometrics, San Diego, CA) were cut to length and tested for light transmission prior to implantation. Prior to behavioral testing, mice were habituated to patch cord tethering daily for 3 days. Using a Neurophotometrics FP3001 system (Neurophotometrics, San Diego, CA), alternating pulses of 470 nm and 415 nm LED light (~50 μW, 40 Hz) were bandpass filtered and focused onto a branching patch cord (Doric, Quebec City, Quebec) by a 20x objective lens. A custom-built Arduino was used to time-lock recordings to behavior. Custom MATLAB scripts were used for all analysis. Background fluorescence was subtracted from 415 and 470 traces, then traces were low-pass filtered at 5 Hz and fit to a biexponential curve. Curve fit was subtracted from each trace to correct for baseline drift. dF/F for 415 and 470 traces were calculated (raw signal-fitted signal)/(fitted signal), and traces were z-scored. The 415 signal was fit to the 470 signal using non-negative robust linear regression to correct for motion artifacts (*Martianova et al., 2019*). Event analysis: Events were identified from processed traces using a custom MATLAB script and were defined as peaks greater than 2* the mean absolute deviation of baseline (first minute of each recording). Peak timing was then aligned with

freezing data to determine whether each event occurred during freezing or mobility. Analysis code is available at https://github.com/oliviahon/FiberPho (*Hon, 2022*).

## 1-Photon calcium imaging

Prior to the start of behavioral testing, mice were habituated to miniature head-mounted microendoscope attachment daily for 3 days. For all experiments, 470 LED lights were delivered at 0.9–1.1 mW, and data were recorded at 20 Hz using an Inscopix nVoke imaging system (Inscopix, Mountain View, CA). Recordings were time-locked to behavior hardware via TTL. Data were analyzed using Inscopix Data Processing Software. Briefly, data were spatially downsampled by a factor of 4 and temporally downsampled by a factor of 2 to reduce computational load. Videos were then spatial bandpass filtered, motion corrected, and normalized with dF/F to prepare the video for automated cell detection using PCA-ICA. Detected regions of interest were then manually accepted or rejected based on signal quality, and overlapping regions were discarded. Events were then identified as events with a minimum peak of 3* the median absolute deviation and minimum decay time of 0.2 s. Traces were then further processed using a custom MATLAB script. Briefly, traces were z-scored, and peri-event plots were generated. Responses to tone and shock were determined for each trial for each cell using a Wilcoxon rank-sum test comparing the 2 s window before and after stimulus. For GCaMP recordings in BNST[CRF], GCaMP8m AAV1-syn-flex-GCaMP8m (Addgene) was diluted 1:1 in sterile PBS and injected unilaterally into the BNST.

## Automated behavior tracking

Behavior videos were tracked using DeepLabCut software that uses deep neural networks to estimate mouse position (*Mathis et al., 2018*). Behavior was then scored by analyzing DLC tracking output using SimBA (*Nilsson et al., 2020*), a machine learning platform that can be trained to identify behaviors. A classifier for freezing behavior was created in-house. A custom MATLAB script (Mathworks, Natick, Massachusetts) was used to determine freezing bout start and end times, and to calculate % freezing during various behavior epochs.

## Histology

To verify virus, fiber, and GRIN lens placement, mice were transcardially perfused with 30 ml each of phosphate-buffered saline and 4% paraformaldehyde. Brains were post-fixed in 4% PFA for 24 hr after extraction, then sectioned on a vibratome. GCaMP fluorescence was amplified using immunohistochemistry. Specifically, we used an anti-GFP antibody (Aves Labs GFP1020) and an anti-chicken 488 secondary (Jackson Labs). Mice with off-target virus expression, fiber, or lens placement were excluded from analysis.

## Statistics

All data were analyzed for statistical significance using Prism software (GraphPad Prism 9). No statistical methods were used to predetermine sample sizes. All behavioral assays were repeated in a minimum of two cohorts with similar replication of results. Littermates were randomly assigned to experimental groups and mice were tested in random order.

---

## Additional information

### Competing interests

Waylin Yu: Affiliated with Inscopix (now bruker). The other authors declare that no competing interests exist.

### Funding

| Funder | Grant reference number | Author |
| --- | --- | --- |
| National Institute on Alcohol Abuse and Alcoholism | AA019454 | Thomas L Kash |

| Funder | Grant reference number | Author |
|---|---|---|

The funders had no role in study design, data collection and interpretation, or the decision to submit the work for publication.

## Author contributions

Olivia J Hon, Conceptualization, Data curation, Formal analysis, Investigation, Methodology, Writing – original draft, Writing – review and editing; Sofia Neira, Formal analysis, Writing – review and editing; Meghan E Flanigan, Data curation, Formal analysis, Investigation, Writing – original draft; Alison V Roland, Conceptualization, Data curation, Formal analysis, Investigation, Writing – original draft; Christina M Caira, Formal analysis, Investigation, Writing – original draft; Tori Sides, Shannon L D'Ambrosio, Sophia I Lee, Yolanda Simpson, Michelle C Buccini, Investigation; Samantha Machinski, Waylin Yu, Formal analysis, Investigation; Kristen M Boyt, Data curation, Investigation; Thomas L Kash, Conceptualization, Resources, Data curation, Supervision, Funding acquisition, Methodology, Writing – original draft, Project administration, Writing – review and editing

## Author ORCIDs

Thomas L Kash (iD) https://orcid.org/0000-0002-4747-4495

## Ethics

All experiments were conducted in accordance with the University of North Carolina at Chapel Hill's Institutional Animal Care Use Committee's guidelines under protocols 19-078 and 22-015.

Reviewer #1 (Public review): https://doi.org/10.7554/eLife.89189.3.sa1
Reviewer #2 (Public review): https://doi.org/10.7554/eLife.89189.3.sa2
Reviewer #3 (Public review): https://doi.org/10.7554/eLife.89189.3.sa3
Author response https://doi.org/10.7554/eLife.89189.3.sa4

# Additional files

## Supplementary files

MDAR checklist

## Data availability

Source data for all figures contain the numerical data used to generate the figures.

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
