## [Editor Report · eLife Assessment]

This **valuable** study advances understanding of how corticotrophin releasing factor in the bed nucleus of the stria terminalis regulates sustained and phasic fear and how this differs between sexes. The evidence is **convincing** and based on state-of-the-art techniques. The work will be of interest to neuroscientists studying the biological basis of fear processing.

---

## [Referee Report · Reviewer #1 (Public review)]

The aim of this study is to test the overarching hypothesis that plasticity in BNST CRF neurons drives distinct behavioral responses to unpredictable threat in males and females. The manuscript provides solid evidence for a sex-specific role for CRF-expressing neurons in the BNST in unpredictable aversive conditioning and subsequent hypervigilance across sexes. As the authors note, this is an important question given the high prevalence of sex differences in stress-related disorders, like PTSD, and the role of hypervigilance and avoidance behaviors in these conditions. The study includes in vivo manipulation, bulk calcium imaging, and cellular resolution calcium imaging, which yield important insights into cell-type specific activity patterns. A major strength of this manuscript is the inclusion of both males and females and attention to possible behavioral and neurobiological differences between them throughout.

---

## [Referee Report · Reviewer #2 (Public review)]

This study examined the role of CRF neurons in the BNST in both phasic and sustained fear in males and females. The authors first established a differential fear paradigm whereby shocks were consistently paired with tones (Full) or only paired with tones 50% of the time (Part), or controls who were exposed to only tones with no shocks. Recall tests established that both Full and Part conditioned male and female mice froze to the tones, with no difference between the paradigms. Additional studies using the NSF and startle test, established that neither fear paradigm produced behavioral changes in the NSF test, suggesting that these fear paradigms do not result in an increase in anxiety-like behavior. Part fear conditioning, but not Full, did enhance startle responses in males but not females, suggesting that this fear paradigm did produce sustained increases in hypervigilance in males exclusively. Photometry studies found that while undifferentiated BNST neurons all responded to shock itself, only Full conditioning in males lead to a progressive enhancement of the magnitude of this response. BNST neurons in males, but not females, were also responsive to tone onset in both fear paradigms, but only in Full fear did the magnitude of this response increase across training. Knockdown of CRF from the BNST had no effect on fear learning in males or females, nor any effect in males on fear recall in either paradigm, but in females enhanced both baseline and tone-induced freezing only in Part fear group. When looking at anxiety following fear training, it was found in males that CRF knockdown modulated anxiety in Part fear trained animals and amplified startle in Full trained males but had no effect in either test in females. Using 1P imaging, it was found that CRF neurons in the BNST generally decline in activity across both conditioning and recall trials, with some subtle sex differences emerging in the Part fear trained animals in that in females BNST CRF neurons were inhibited after both shock and omission trials but in males this only occurred after shock and not omission trials. In recall trials, CRF BNST neuron activity remained higher in Part conditioned mice relative to Full conditioned mice.

Overall, this is a very detailed and complex study that incorporates both differing fear training paradigms and males and females, as well as a suite of both state-of-the-art imaging techniques and gene knockdown approaches to isolate the role and contributions of CRF neurons in the BNST to these behavioral phenomena. The strengths of this study come from the thorough approach that the authors have taken, which in turn helped to elucidate nuanced and sex specific roles of these neurons in the BNST to differing aspects of phasic and sustained fear. More so, the methods employed provide a strong degree of cellular resolution for CRF neurons in the BNST. In general, the conclusions appropriately follow the data, although the authors do tend to minimize some of the inconsistencies across studies, although this has now been addressed to some degree. The discussion has also been improved to now address some of the inconsistencies in the data head on. Discussion of a few other points is below:

- Given the focus on CRF neurons in the BNST, it was unclear why the photometry studies were performed in undifferentiated BNST neurons as opposed to CRF neurons specifically, although the authors have now explained this in better depth making this clearer to the reader.

- The CRF KD studies are interesting, but it remains speculative as to whether these effects are mediated locally in the BNST or due to CRF signaling at downstream targets. As the literature on local pharmacological manipulation of CRF signaling within the BNST seems to be largely performed in males, the addition of pharmacological studies here would benefit this to help to resolve if these changes are indeed mediated by local impairments in CRF release within the BNST or not. While it is not essential to add these experiments, the authors have addressed this point in the discussion and highlighted studies like this as necessary in future work.

- The authors have addressed the difference between arousal and anxiety by expanding the discussion to include more focus on the behavioral measures. The CRF KD data are still somewhat confusing but better contextualized now. Overall, the manuscript has been improved by the revisions and edits the authors have made.

---

## [Referee Report · Reviewer #3 (Public review)]

Hon et al. investigated the role of BNST CRF signaling in modulating phasic and sustained fear in male and female mice. They found that partial and full fear conditioning had similar effects in both sexes during conditioning and during recall. However, males in the partially reinforced fear conditioning group showed enhanced acoustic startle, compared to the fully reinforced fear conditioning group, an effect not seen in females. Using fiber photometry to record calcium activity in all BNST neurons, the authors show that the BNST was responsive to foot shock in both sexes and both conditioning groups. Shock response increased over the session in males in the fully conditioned fear group, an effect not observed in the partially conditioned fear group. This effect was not observed in females. Additionally, tone onset resulted in increased BNST activity in both male groups, with the tone response increasing over time in the fully conditioned fear group. This effect was less pronounced in females, with partially conditioned females exhibiting a larger BNST response. During recall in males, BNST activity was suppressed below baseline during tone presentations and was significantly greater in the partially conditioned fear group. Both female groups showed an enhanced BNST response to the tone that slowly decayed over time. Next, they knocked CRF in the BNST to examine its effect on fear conditioning, recall and anxiety-like behavior after fear. They found no effect of the knockdown in either sex or group during fear conditioning. During fear recall, BNST CRF knockdown lead to an increase in freezing in only the partially conditioned females. In the anxiety-like behavior tasks, BNST CRF knockdown lead to increased anxiolysis in the partially reinforced fear male, but not in females. Surprisingly, BNST CRF knockdown increased startle response in fully conditioned, but not partially conditioned males. An effect not observed in either female group. In a final set of experiments, the authors single photon calcium imaging to record BNST CRF cell activity during fear conditioning and recall. Approximately, 1/3 of BNST CRF cells were excited by shock in both sexes, with the rest inhibited and no differences were observed between sexes or group during fear conditioning. During recall, BNST CRF activity decreased in both sexes, an effect pronounced in male and female fully conditioned fear groups.

Overall, these data provide novel, intriguing evidence in how BNST CRF neurons may encode phasic and sustained fear differentially in males and females. The experiments were rigorous. My biggest concerns I have regard the interpretations and some conclusions from this data set, which I have stated below.

(1) It was surprising to see minimal and somewhat conflicting behavioral effects due to BNST CRF knockdown. The authors provide a representative image and address this in the conclusion. They mention the role of local vs projection CRF circuits as well as the role of GABA. I don't think those experiments are necessary for this manuscript. However, it may be worthwhile to see through in situ hybridization or IHC, to see BNST CRF levels after both full and partial conditioned fear paradigms. Additionally, it would help to see a quantification of the knockdown of the animals. The authors can add a figure showing deltaF/F changes from control.

(2) Related to the previous point, it was surprising to see an effect of the CRF deletion in the full fear group compared to the partial fear in the acoustic startle task. To strengthen the conclusion about differential recruitment of CRF during phasic and sustained fear, the experiment in my previous point could help elucidate that. Conversely, intra-BNST administration of a CRF antagonist into the BNST before the acoustic startle after both conditioning tasks could also help. Or patch from BNST CRF neurons after the conditioning tasks to measure intrinsic excitability. Not all these experiments are needed to support the conclusion, it's some examples.

(3) In Figure 5 F and K, the authors report data combined for both part and full fear conditioning. Were there any differences between the number of excited or inhibited neurons b/t the conditioning groups? Also, can the authors separate male and female traces in Fig 5 E and P?

(4) Also, regarding the calcium imaging data, what was the average length of a transient induced by shock? Were there any differences between the sexes?

---

## [Author Response]

The following is the authors’ response to the original reviews

**Public Reviews:**

**Reviewer #1 (Public Review):**
The aim of this study is to test the overarching hypothesis that plasticity in BNST CRF neurons drives distinct behavioral responses to unpredictable threat in males and females. The manuscript provides evidence for a possible sex-specific role for CRF-expressing neurons in the BNST in unpredictable aversive conditioning and subsequent hypervigilance across sexes. As the authors note, this is an important question given the high prevalence of sex differences in stress-related disorders, like PTSD, and the role of hypervigilance and avoidance behaviors in these conditions. The study includes in vivo manipulation, bulk calcium imaging, and cellular resolution calcium imaging, which yield important insights into cell-type specific activity patterns. However, it is difficult to generate an overall conclusion from this manuscript, given that many of the results are inconsistent across sexes and across tests and there is an overall lack of converging evidence. For example, partial conditioning yields increased startle in males but not females, yet, CRF KO only increases startle response in males after full conditioning, not partial, and CRF neurons show similar activity patterns between partial and full conditioning across sexes. Further, while the study includes a KO of CRF, it does not directly address the stated aim of assessing whether plasticity in CRF neurons drives the subsequent behavioral effects unpredictable threat.

We appreciate the reviewer’s summary and agree that there is a large amount of complexity to the results, and that it was difficult to generate a simple model/conclusion to summarize our work. This is the unfortunate side effect of looking across both sexes at different conditioning paradigms, however, we believe that it is important to convey this information to the field even without a simple answer. Our data reinforces the very important findings from the Maren and Holmes groups that partial fear is a different process than full fear, and that the BNST plays a differential role here. We have reworded the manuscript to better convey this complexity.

A major strength of this manuscript is the inclusion of both males and females and attention to possible behavioral and neurobiological differences between them throughout. However, to properly assess sex-differences, sex should be included as a factor in ANOVA (e.g. for freezing, startle, and feeding data in Figure 1) to assess whether there is a significant main effect or interaction with sex. If sex is not a statistically significant factor, both sexes should be combined for subsequent analyses. See, Garcia-Sifuentes and Maney, eLife 2021 https://elifesciences.org/articles/70817. There are additional cases where t-tests are used to compare groups when repeated measures ANOVAs would be more appropriate and rigorous.

We agree with the reviewer that this is the more appropriate analysis and have changed the analysis and figures throughout the revised manuscript to better assess sex differences as well as differences between fear conditions.

Additionally, it's unclear whether the two sexes are equally responsive to the shock during conditioning and if this is underlying some of the differences in behavioral and neuronal effects observed. There are some reports that suggest shock sensitivity differs across sexes in rodents, and thus, using a standard shock intensity for both males and females may be confounding effects in this study.

This is a great point. We have conducted appropriate analysis (Sex by Tone Repeated measures two-way ANOVAS for each of the groups: Ctrl, Full, Part) and there are no sex differences in freezing between males and females. The extent of conditioning is not different between the groups suggesting that if there was a difference in shock sensitivity, it is not driving any discernible differences in behavioral performance. However, it is possible that the experience of the shock differs for the animals even in the absence of any measurable behavior.

The data does not rule out that BNST CRF activity is not purely tracking the mobility state of the animal, given that the differences in activity also track with differences in freezing behavior. The data shows an inverse relationship between activity and freezing. This may explain a paradox in the data which is why males show a greater suppression of BNST activity after partial conditioning than full conditioning, if that activity is suspected to drive the increased anxiety-like response. Perhaps it reflects that activity is significantly suppressed at the end of the conditioning session because animals are likely to be continuously freezing after repeated shock presentations in that context. It would also explain why there is less of a suppression in activity over the course of the recall session, because there is less freezing as well during recall compared with conditioning.

While it is possible that the BNST may be tracking activity, we believe it is not purely tracking mobility state. For instance, while freezing increases across tone exposures in Part fear regardless of sex, males show an increase while females show a reduction in BNST response during tone 5 (Fig 2K). The data the reviewer refers to showing the inverse relationship with BNST activity and freezing would have suggested the opposite response if it were purely tracking the mobility state of the animal. This is also the case with BNST^CRF^ activity to first and last tone during recall. Despite the suppression of activity over the course of recall (Fig 5K), we see an increase in BNST^CRF^ tone response when comparing tone 1 and 6 in males and a decrease in females (Fig 6M), again suggesting the BNST is responding to more than just activity.

A mechanistic hypothesis linking BNST CRF neurons, the behavioral effects observed after fear conditioning, and manipulation of CRF itself are not clearly addressed here.

We disagree with this assertion. The data suggests a model in which males respond with increased arousal and Part fear males show persistent activation of the BNST and BNST^CRF^ neurons during fear conditioning and recall while female Part fear mice show the opposite response. This female response differs from what the field believes to be the role of the BNST in sustained fear. Additionally, we show that CRF knockdown is not involved in fear differentiation or fear expression in males, while it enhances fear learning and recall in females. We have reworded the manuscript to highlight these novel findings.

**Reviewer #2 (Public Review):**
This study examined the role of CRF neurons in the BNST in both phasic and sustained fear in males and females. The authors first established a differential fear paradigm whereby shocks were consistently paired with tones (Full) or only paired with tones 50% of the time (Part), or controls who were exposed to only tones with no shocks. Recall tests established that both Full and Part conditioned male and female mice froze to the tones, with no difference between the paradigms. Additional studies using the NSF and startle test, established that neither fear paradigm produced behavioral changes in the NSF test, suggesting that these fear paradigms do not result in an increase in anxiety-like behavior. Part fear conditioning, but not Full, did enhance startle responses in males but not females, suggesting that this fear paradigm did produce sustained increases in hypervigilance in males exclusively.

Thank you for this clear summary of the behavioral work.

Photometry studies found that while undifferentiated BNST neurons all responded to shock itself, only Full conditioning in males lead to a progressive enhancement of the magnitude of this response. BNST neurons in males, but not females, were also responsive to tone onset in both fear paradigms, but only in Full fear did the magnitude of this response increase across training. Knockdown of CRF from the BNST had no effect on fear learning in males or females, nor any effect in males on fear recall in either paradigm, but in females enhanced both baseline and tone-induced freezing only in Part fear group. When looking at anxiety following fear training, it was found in males that CRF knockdown modulated anxiety in Part fear trained animals and amplified startle in Fully trained males but had no effect in either test in females. Using 1P imaging, it was found that CRF neurons in the BNST generally decline in activity across both conditioning and recall trials, with some subtle sex differences emerging in the Part fear trained animals in that in females BNST CRF neurons were inhibited after both shock and omission trials but in males this only occurred after shock and not omission trials. In recall trials, CRF BNST neuron activity remained higher in Part conditioned mice relative to Full conditioned mice.Overall, this is a very detailed and complex study that incorporates both differing fear training paradigms and males and females, as well as a suite of both state of the art imaging techniques and gene knockdown approaches to isolate the role and contributions of CRF neurons in the BNST to these behavioral phenomena. The strengths of this study come from the thorough approach that the authors have taken, which in turn helped to elucidate nuanced and sex specific roles of these neurons in the BNST to differing aspects of phasic and sustained fear. More so, the methods employed provide a strong degree of cellular resolution for CRF neurons in the BNST. In general, the conclusions appropriately follow the data, although the authors do tend to minimize some of the inconsistencies across studies (discussed in more depth below), which could be better addressed through discussion of these in greater depth. As such, the primary weakness of this manuscript comes largely from the discussion and interpretation of mixed findings without a level of detail and nuance that reflects the complexity, and somewhat inconsistency, across the studies. These points are detailed below:- Given the focus on CRF neurons in the BNST, it is unclear why the photometry studies were performed in undifferentiated BNST neurons as opposed to CRF neurons specifically (although this is addressed, to some degree, subsequently with the 1P studies in CRF neurons directly). This does limit the continuity of the data from the photometry studies to the subsequent knockdown and 1P imaging studies. The authors should address the rationale for this approach so it is clear why they have moved from broader to more refined approaches.

The reviewer raises a good point. We did some preliminary photometry studies with BNST CRF neurons and found that there was poor time locked signal. We reasoned that this was due to the heterogeneity of the cell activity, as we saw in our previous publication (Yu et al). Because of this, we moved to the 1p imaging work in place of continued BNST CRF photometry. We have also reworded the manuscript to better discuss the complexities and inconsistencies in findings across the studies.

- The CRF KD studies are interesting, but it remains speculative as to whether these effects are mediated locally in the BNST or due to CRF signaling at downstream targets. As the literature on local pharmacological manipulation of CRF signaling within the BNST seems to be largely performed in males, the addition of pharmacological studies here would benefit this to help to resolve if these changes are indeed mediated by local impairments in CRF release within the BNST or not. While it is not essential to add these experiments, the manuscript would benefit from a more clear description of what pharmacological studies could be performed to resolve this issue.

We agree with the reviewer that the addition of this experiment would be highly informative for differentiating the role of CRF in the BNST. This is something that will need to be considered moving forward and we have added this as a point of discussion.

- While I can appreciate the authors perspective, I think it is more appropriate to state that startle correlates with anxiety as opposed to outright stating that startle IS anxiety. Anxiety by definition is a behavioral cluster involving many outputs, of which avoidance behavior is key. Startle, like autonomic activation, correlates with anxiety but is not the same thing as a behavioral state of anxiety (particularly when the startle response dissociates from behavior in the NSF test, which more directly tests avoidance and apprehension). Throughout the manuscript the use of anxiety or vigilance to describe startle becomes interchangeable, but then the authors also dissociate these two, such as in the first paragraph of the discussion when stating that the Part fear paradigm produces hypervigilance in males without influencing fear or anxiety-like behaviors. The manuscript would benefit from harmonization of the language used to operationally define these behaviors and my recommendation would be to remain consistent with the description that startle represents hypervigilance and not anxiety, per se.

The reviewer raises an excellent point, we have clarified in the revised manuscript.

- The interpretation of the anxiety data following CRF KD is somewhat confusing. First, while the authors found no effect of fear training on behavior in the NSF test in the initial studies, now they do, however somewhat contradictory to what one would expect they found that Full fear trained males had reduced latency to feed (indicative of an anxiolytic response), which was unaltered by CRF KD, but in Part fear (which appeared to have no effect on its own in the NSF test), KD of CRF in these animals produced an anxiolytic effect. Given that the Part fear group was no different from control here it is difficult to interpret these data as now CRF KD does reduce latency to feed in this group, suggesting that removal of CRF now somehow conveys an anxiolytic response for Part fear animals. In the discussion the authors refer to this outcome as CRF KD "normalizing" the behavior in the NSF test of Part fear conditioned animals as now it parallels what is seen after Full fear, but given that the Part fear animals with GFP were no different then controls (and neither of these fear training paradigms produced any effect in the NSF test in the first arm of studies), it seems inappropriate to refer to this as "normalization" as it is unclear how this is now normalized. Given the complexity of these behavioral data, some greater depth in the discussion is required to put these data in context and describe the nuance of these outcomes, in particular a discussion of possible experimental factors between the initial behavioral studies and those in the CRF KD arm that could explain the discrepancy in the NSF test would be good (such as the inclusion of surgery, or other factors that may have differed between these experiments). These behavioral outcomes are even more complex given that the opposite effect was found in startle whereby CRF KD amplified startle in Full trained animals. As such, this portion of the discussion requires some reworking to more adequately address the complexity of these behavioral findings.

The reviewer raises a good point, and we agree that there are many inconsistencies in the behaviors. We believe it is still good to show these results but have expanded the manuscript on potential reasons for these behavioral inconsistencies.

**Reviewer #3 (Public Review):**
Hon et al. investigated the role of BNST CRF signaling in modulating phasic and sustained fear in male and female mice. They found that partial and full fear conditioning had similar effects in both sexes during conditioning and during recall. However, males in the partially reinforced fear conditioning group showed enhanced acoustic startle, compared to the fully reinforced fear conditioning group, an effect not seen in females. Using fiber photometry to record calcium activity in all BNST neurons, the authors show that the BNST was responsive to foot shock in both sexes and both conditioning groups. Shock response increased over the session in males in the fully conditioned fear group, an effect not observed in the partially conditioned fear group. This effect was not observed in females. Additionally, tone onset resulted in increased BNST activity in both male groups, with the tone response increasing over time in the fully conditioned fear group. This effect was less pronounced in females, with partially conditioned females exhibiting a larger BNST response. During recall in males, BNST activity was suppressed below baseline during tone presentations and was significantly greater in the partially conditioned fear group. Both female groups showed an enhanced BNST response to the tone that slowly decayed over time. Next, they knocked CRF in the BNST to examine its effect on fear conditioning, recall and anxiety-like behavior after fear. They found no effect of the knockdown in either sex or group during fear conditioning. During fear recall, BNST CRF knockdown lead to an increase in freezing in only the partially conditioned females. In the anxiety-like behavior tasks, BNST CRF knockdown lead to increased anxiolysis in the partially reinforced fear male, but not in females. Surprisingly, BNST CRF knockdown increased startle response in fully conditioned, but not partially conditioned males. An effect not observed in either female group. In a final set of experiments, the authors single photon calcium imaging to record BNST CRF cell activity during fear conditioning and recall. Approximately, 1/3 of BNST CRF cells were excited by shock in both sexes, with the rest inhibited and no differences were observed between sexes or group during fear conditioning. During recall, BNST CRF activity decreased in both sexes, an effect pronounced in male and female fully conditioned fear groups.Overall, these data provide novel, intriguing evidence in how BNST CRF neurons may encode phasic and sustained fear differentially in males and females. The experiments were rigorous.

We thank you for this positive review of our manuscript.

**Recommendations for the authors:**

**Reviewer #1 (Recommendations For The Authors):**
There are several graphs representing different analyses of (presumably) the same group of subjects, but which have different N/group. For example, in Figure 2:(1) Fig 2P seems to have n=10 in Part Male group (Peak), but 2Q only has n=9 in Part Male group (AUC)(2) Fig 2S seems to have n=10 in Part Female group (Peak), but 2T only has n=7 in Part Female group (AUC)(3) Fig 2G (Tone Resp) has n=6 Full Males but 2F (Tone Resp), 2H (Shock Resp), and 2I (Shock Resp) have n=7 Full Males(4) Fig 2K (Tone Resp) has n=7 Full Females but 2L (Tone Resp), 2M (Shock Resp), and 2N (Shock Resp) have n=8 Full Females(5) Fig 2L (Tone Resp) has n=9 Part Females but 2K (Tone Resp), 2M (Shock Resp), and 2N (Shock Resp) have n=10 Part FemalesIt's possible that this is just due to overlapping individual data points which are made harder to see due to the low resolution of the figures. If so, this can be easily rectified. However, there may also be subjects missing from some analyses which must be clarified or corrected.

We thank you for catching these. We have gone through and fixed any issues with data points and have added statistics and exclusions in datasets to figure legends to further explain inconsistencies.

Regarding statistical tests:(2) Data in Figs 2G and 2I should be analyzed using a two-way RM ANOVA.

We have now included sex as a factor in most of our analysis and are now using appropriate statistical tests.

(3) Data in Fig 3K should be analyzed using a two-way RM ANOVA.

We are now using appropriate statistical tests.

Calcium activity in response to the shock during conditioning and in response to the tone during recall should be included in Figure 5. Given partial and full animals also receive unequal presentations of the cue, it would be useful to see the effects trial by trial or normalized to the first 3 presentations only.

The reviewer raises a great point. We have changed this figure and have now added the response to shock and tones. Since we are most interested in the difference between sustained and phasic fear, we decided to compare tone 3 in Full fear and tone 4 in Part fear, which differ in the ambiguity of their cue and only have one tone difference.

Histology maps should be included for all experiments depicting viral spread and implant location for all animals, in addition to the included representative histology images. These can be placed in the supplement.

We agree this is helpful. While we have confirmed all of the experiments are hits, the tissue is no longer in condition for this analysis.

Referring to the quantification of peaks in fiber photometry and cellular resolution calcium imaging data as "spikes" is a bit misleading given the inexact relationship between GCAMP sensor dynamics/calcium binding and neuronal action potentials, perhaps calling it "event" frequency would be more clear.

We have changed the references of spikes to events as suggested.

The legend for Figure 2S is mislabeled as A.

Thank you for catching this mistake, it has been fixed.

The methods refer to CRFR1 fl/fl animals but it seems no experiments used these animals, only CRF fl/fl.

We have fixed this, thank you.

**Reviewer #2 (Recommendations For The Authors):**
As stated in the public review, while I think the addition of local pharmacological studies blocking CRF1 and 2 receptors in the BNST in both males and females, done under the same conditions as all of the other testing herein, would help to resolve some of the speculation of interpreting the CRF KD data, I dont think these studies are essential to do, but it would be good for the authors to more explicitly state what studies could be done and how they could facilitate interpretation of these data.

Thank you for this suggestion. We have added this discussion into the manuscript.

Asides from this, my other recommendations for the authors are to more clearly address the discrepancies in behavioral outcomes across studies and explicitly describe their rationale for the sequence of experiments performed and to harmonize their operationalization of how they define anxiety.

Again, we appreciate these great suggestions. We have added more discussion on the behavioral discrepancies as well as rationale for the experiments. We have also changed the wording to remain consistent that the NSF test relates to anxiety and the Startle test relates to vigilance.

- In Figure 2, Panel S is listed as Panel A in the caption and should be corrected.

Thank you for catching this mistake, we have fixed it.

**Reviewer #3 (Recommendations For The Authors):**
My biggest concerns I have regard the interpretations and some conclusions from this data set, which I have stated below.(1) It was surprising to see minimal and somewhat conflicting behavioral effects due to BNST CRF knockdown. The authors provide a representative image and address this in the conclusion. They mention the role of local vs projection CRF circuits as well as the role of GABA. I don't think those experiments are necessary for this manuscript. However, it may be worthwhile to see through in situ hybridization or IHC, to see BNST CRF levels after both full and partial conditioned fear paradigms. Additionally, it would help to see a quantification of the knockdown of the animals.

Thank you for these great suggestions. We will consider these for future experiments. We piloted out some CRF sensor experiments to probe this, but it was unclear if the signal to noise for the sensor was sufficient. We hope to do more of this in the future if we ever manage to get funding for this work.

The authors can add a figure showing deltaF/F changes from control.

We did not have control mice in these in-vivo experiments Our main interests lie in understanding the differences in Full and Part Fear conditioning paradigms specifically.

(2) Related to the previous point, it was surprising to see an effect of the CRF deletion in the full fear group compared to the partial fear in the acoustic startle task. To strengthen the conclusion about differential recruitment of CRF during phasic and sustained fear, the experiment in my previous point could help elucidate that. Conversely, intra-BNST administration of a CRF antagonist into the BNST before the acoustic startle after both conditioning tasks could also help. Or patch from BNST CRF neurons after the conditioning tasks to measure intrinsic excitability. Not all these experiments are needed to support the conclusion, it's some examples.

We thank the reviewer for these suggestions and agree that these are important experiments. We will consider this in future experiments exploring the role of BNST CRF in fear conditioning.

(3) In Figure 5 F and K, the authors report data combined for both part and full fear conditioning. Were there any differences between the number of excited or inhibited neurons b/t the conditioning groups?

We are only looking at the first shock exposure in these figures. These were combined because the first tone and shock exposure is identical in Full and Part fear conditioning. Differences in these behavioral paradigms emerge after Tone 3 exposure, where Part fear does not receive a shock while Full fear does.

Also, can the authors separate male and female traces in Fig 5 E and P?

Traces in Fig E are from females only. We did not include male traces because males and females had identical responses to first shock, and we felt only one trace was needed as an example. Traces in Figure P are from males. We did not show female traces because females did not show differential effects from baseline to end.

(4) Also, regarding the calcium imaging data, what was the average length of a transient induced by shock? Were there any differences between the sexes?

We have many cells in each condition, and the length of traces after shock were all different and hard to quantify, as for example, sometimes cells were active before shock and thus trace length would be difficult to quantify. Therefore, to keep consistency and reduce ambiguity regarding trace lengths, we focused on keeping the time consistent across mice and focused on the 10 second window post shock to be consistent across conditions.